# DARK MINER: DEFEND AGAINST UNSAFE GENERATION FOR TEXT-TO-IMAGE DIFFUSION MODELS

## ABSTRACT

Text-to-image diffusion models have been demonstrated with unsafe generation due to unfiltered large-scale training data, such as violent, sexual, and shocking images, necessitating the erasure of unsafe concepts. Most existing methods focus on modifying the generation probabilities conditioned on the texts containing unsafe descriptions. However, they fail to guarantee safe generation for unseen texts in the training phase, especially for the prompts from adversarial attacks. In this paper, we re-analyze the erasure task and point out that existing methods cannot guarantee the minimization of the total probabilities of unsafe generation. To tackle this problem, we propose Dark Miner. It entails a recurring three-stage process that comprises mining, verifying, and circumventing. It greedily mines embeddings with maximum generation probabilities of unsafe concepts and reduces unsafe generation more effectively. In the experiments, we evaluate its performance on two inappropriate concepts, two objects, and two styles. Compared with 6 previous state-of-the-art methods, our method achieves better erasure and defense results in most cases, especially under 4 state-of-the-art attacks, while preserving the model's native generation capability. Our code can be found in Supplementary Material and will be available on GitHub.

**Warning:** This paper may contain disturbing, distressing, or offensive content.

## 1 INTRODUCTION

Recently, the rapid development of text-to-image diffusion models (Gal et al., 2022; Mou et al., 2024; Nichol et al., 2022; Rombach et al., 2022; Ruiz et al., 2023; Saharia et al., 2022), such as Stable Diffusion (Rombach et al., 2022), pushes the performance of high-fidelity controllable image generation to a new height. These models are trained on large-scale text-image pairs and learn to capture semantic connections between texts and images. However, everything has two sides. The training data is randomly crawled from the Internet without being filtered due to its large scale. It results in the inclusion of unsafe content such as violent and sexual images, thus bringing the risk of unsafe generation of the models (Qu et al., 2023). The generation of unsafe concepts affects social harmony and stability, hindering the safe use of these generative models.

Researchers have explored different methods to prevent a trained diffusion model from unsafe generation. These methods can be broadly classified into two categories. The first category includes the training-free methods, such as Safe Latent Diffusion (Schramowski et al., 2023) which defines inappropriate concept sets and modifies their generation guidance. The second category includes the fine-tuning-based methods, which align the generation distributions of unsafe texts to anchor texts by fine-tuning model weights. Some examples include Gandikota et al. (2023; 2024); Kumari et al. (2023). Other works like Forget-Me-Not (Zhang et al., 2023a) suppress activation of unsafe content in attention maps, while the works Bui et al. (2024); Huang et al. (2023) introduce learnable prompts and adversarial training for better erasure performance. SalUn (Fan et al., 2024), on the other hand, proposes to fine-tune a model based on the saliency of model weights with the output.

The existing studies mainly focus on modifying the generation distributions conditioned on the texts containing unsafe descriptions (Bui et al., 2024; Gandikota et al., 2023; 2024; Huang et al., 2023; Kumari et al., 2023; Schramowski et al., 2023; Zhang et al., 2023a). Therefore, how to identify these unsafe texts becomes a key point. These methods use prompt templates like *a * photo* (Bui et al., 2024; Fan et al., 2024; Gandikota et al., 2024; Schramowski et al., 2023; Zhang et al., 2023a) or

acquire a large number of relevant texts from Large Language Models or datasets (Gandikota et al., 2023; Huang et al., 2023; Kumari et al., 2023). While these solutions can ensure the safety of the texts involved in the training, they cannot guarantee the safety of unseen texts. On the one hand, there are still unsafe texts that cannot be covered beforehand. On the other hand, even if a given text does not explicitly suggest unsafe concepts, the unsafe knowledge of the models can still lead to unsafe images. This issue makes the models not only unsafe but also highly vulnerable to malicious attacks (Chin et al., 2023; Pham et al., 2023; Tsai et al., 2024; Zhang et al., 2023b).

To tackle this challenge, we conduct an analysis of the erasure task. We point out that the objective of the task is to minimize the overall likelihood of generating unsafe content, whereas current methods solely focus on a portion of it. Ideally, we would devise a comprehensive set encompassing all texts related to unsafe images, but such an endeavor remains impractical. To approximate it in an effective way, we propose a greedy method that circumvents unsafe generation from a global perspective. Specifically, we propose **D**ark **M**iner for text-to-image **D**iffusion **M**odels, or DM$^2$. The method is a recurring three-stage process including mining, verifying, and circumventing. In the mining stage, DM$^2$ learns a text embedding with the highest likelihood of generating unsafe concepts. In the verifying stage, DM$^2$ assesses whether the embedding can lead to unsafe concepts, leveraging reference images and anchor images as benchmarks. If the verification is successful, the circumventing stage commences, where DM$^2$ fine-tunes the models to modify the conditional generation probabilities to the one conditioned on an anchor text, ultimately returning to the mining stage. Through the above process, it continuously reduces a tight upper bound on the overall likelihood of unsafe generation, thus realizing a reduction in the overall likelihood. In the experiments, we compare its performance with 6 state-of-the-art methods in erasing 6 concepts. The concepts include the inappropriateness (nudity and violence), the objects (church and French horn), and the styles (Van Gogh and crayon). The performance against 4 state-of-the-art attacks is reported as well. The results show that DM$^2$ achieves the best erasure performance and the best anti-attack performance while preserving the ability of generations conditioned on general texts. In summary, the contributions of this paper are as follows.

- Based on the total probabilities, we analyze the reason why existing methods cannot completely erase concepts for text-to-image diffusion models and are vulnerable to attacks.

- To tackle this challenge, we propose DM$^2$. It involves a recurring three-stage process, mining optimal embeddings related to unsafe concepts and circumventing them after verification.

- DM$^2$ is evaluated with 6 state-of-the-art methods. We evaluate the erasure performance on 6 concepts and the anti-attack performance when faced with 4 state-of-the-art attacks. DM$^2$ achieves the best results while preserving the native generative capabilities.

## 2 RELATED WORK

### 2.1 TEXT-TO-IMAGE DIFFUSION MODELS

Based on the Markov forward and backward diffusion process, diffusion models (Ho et al., 2020; Sohl-Dickstein et al., 2015) train a noise estimator $\epsilon_\theta(x_t|t)$, which is a U-Net architecture (Ronneberger et al., 2015), to estimate and remove noises from the sampled Gaussian noises step-by-step. Different from the random generation of images, text-to-image diffusion models (Gal et al., 2022; Mou et al., 2024; Nichol et al., 2022; Rombach et al., 2022; Ruiz et al., 2023; Saharia et al., 2022) achieve text-guided image generation. Specifically, they use a text encoder to encode a given text $c$ into features. Some cross-attention modules are inserted between the middle layers of the diffusion models, and regard the text features as keys and the image features as queries and values. In this way, a diffusion model becomes a noise estimator $\epsilon_\theta(x_t|t, c)$ conditioned on the text $c$. The models are trained by the following objective:

$$\mathbb{E}_{(x,c)\sim\mathcal{D}, \epsilon\in\mathcal{N}(0,\mathbf{I}), t\in\mathcal{U}(0,T)} \left[ ||\epsilon - \epsilon_\theta(x_t|t,c)||_2^2 \right] \tag{1}$$

where $(x, c)$ is the image-text pair from the dataset $\mathcal{D}$, $\epsilon$ is the random Gaussian noise, $t$ is the time step sampled from the uniform distribution $\mathcal{U}(0, T)$, and $x_t = \sqrt{\overline{\alpha}_t}x + \sqrt{1 - \overline{\alpha}_t}\epsilon$ where $\overline{\alpha}_t = \prod_{i=1}^{t} \alpha_i$ and $\alpha_t(t = T, T - 1, ..., 0)$ are the scheduled coefficients. Text-to-image diffusion models learn to fit a conditional probability distribution $p_\theta(x|c)$ from a real data distribution $q(x|c)$.

## 2.2 CONCEPT ERASURE

The large-scale datasets for training text-to-image diffusion models, usually crawled from the Internet, contain unsafe or unexpected images. For example, LAION-5B (Schuhmann et al., 2022), which is the training set of Stable Diffusion (Rombach et al., 2022), has many violent, bloody, or sexual images. It leads to unsafe image generation. Many methods have been proposed to erase unsafe concepts from trained diffusion models. These methods can be classified into two categories. The first is training-free, such as Safe Latent Diffusion (Schramowski et al., 2023). They prevent unsafe generation by interfering with the generation processes or results. The second category requires the updates of model weights. Erasing Stable Diffusion (Gandikota et al., 2023) and Concept Ablating (Kumari et al., 2023) modify the generation distributions conditioned on gathered unsafe texts via fine-tuning attention weights. Forget-Me-Not (Zhang et al., 2023a) suppresses the activation of attention maps associated with to-be-erased concepts. Bui et al. (2024) and Huang et al. (2023) have introduced learnable prompts in the fine-tuning. Unlike these fine-tuning methods, SalUn (Fan et al., 2024) proposes to analyze specific model weights related to concepts. Without any training, Unified Concept Editing (Gandikota et al., 2024) proposes a closed-form minimum solution for the parameters of attention layers. These methods except for SalUn erase unsafe concepts according to limited given prompts. While SalUn differs from them with better erasure performance, it sacrifices the generative performance, leading to a significant drop in the generative performance. Contrary to existing methods, Our method greedily mines embeddings with the highest likelihood of unsafe generation in an iterative manner, reducing the overall probabilities of unsafe concepts more effectively.

## 2.3 ERASURE ATTACKS

Some researchers design attacking methods to render the erasure methods ineffective. They search for adversarial prompts to lead models to generate unsafe images. Circumventing Concept Erasure (Pham et al., 2023), Prompting4Debugging (Chin et al., 2023), and UnlearnDiff (Zhang et al., 2023b) are three attacking methods that use sanitized diffusion models to optimize safety-evasive prompts. Different from them, Ring-A-Bell (Tsai et al., 2024) finds adversarial prompts by the genetic algorithm and CLIP (Radford et al., 2021). It provides a model-agnostic adversarial prompt searching tool. Experiments reveal that most existing erasing methods cannot effectively defend against these attacks, exposing their incompleteness in erasing unsafe concepts.

# 3 METHODS

## 3.1 ANALYSIS OF THE ERASURE TASK

Previous studies formulate the task of unsafe concept erasure as a modification of generation distributions for known texts. This subsection re-analyzes the task. Denote the unsafe concept to be erased as $e$, the generated image containing the concept $e$ as $x_e$, and the probability of generating $x_e$ as $p_\theta(x_e)$ where $\theta$ is the parameters of text-to-image diffusion models. Here, $x_e$ does not refer to any specific image, but rather to images that contain the concept $e$. We think that the goal of the task is to prevent the models from generating $x_e$, i.e. $\min p_\theta(x_e)$. We also define an open set $\mathcal{C}$, which contains all texts $c$ that may be input as a condition for diffusion models. According to the Total Probability, the probability of generating $x_e$ can be expressed as the following form:

$$p_\theta(x_e) = \sum_{c \in \mathcal{C}} p(c) p_\theta(x_e|c). \tag{2}$$

Here, $p(c)$ denotes the prior probability of $c$ and $\sum_{c \in \mathcal{C}} p(c) = 1$. Eq.2 shows that minimizing the likelihood of generating $x_e$ requires reducing the probability of unsafe generation conditioned on each possible text $c$. However, most existing methods, such as Gandikota et al. (2023; 2024); Kumari et al. (2023); Schramowski et al. (2023); Zhang et al. (2023a), only focus on a subset $\mathcal{C}'$ of $\mathcal{C}$, i.e. $\min \sum_{c \in \mathcal{C}'} p(c) p_\theta(x_e|c)$. $\mathcal{C}'$ usually contains some predefined prompt templates or collected texts related to $e$. Using these methods, there are still texts that can generate $x_e$, and it is easy to be tricked into generating $x_e$ using various attacking methods.

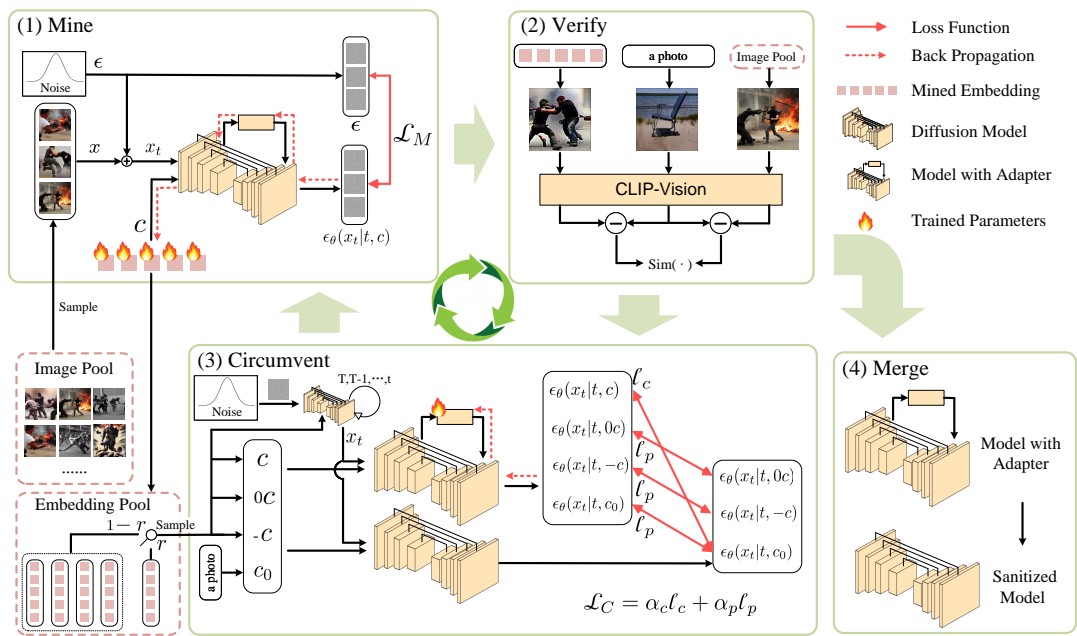

Figure 1: The framework of Dark Miner for text-to-image Diffusion Models ($DM^2$).

## 3.2 DARK MINER

In Sec. 3.1, we point out that the gap between existing methods and the task lies in the difference in the text sets, i.e. $\mathcal{C}'$ and $\mathcal{C}$. However, bridging this gap is not a simple work. $\mathcal{C}$ is an open set and we cannot include all possible texts in the training or inference process. Additionally, an approach that involves all texts may also lead to a significant drop in the generation performance. Revisiting Eq.2, we notice that there is a tight upper bound on it:

$$p_\theta(x_e) = \sum_{c\in\mathcal{C}} p(c)p_\theta(x_e|c) \le \sum_{c\in\mathcal{C}} p(c)M = M. \tag{3}$$

where $M = \max_{c\in\mathcal{C}} p_\theta(x_e|c)$. If and only if $p_\theta(x_e|c) = M(\forall c \in \mathcal{C})$, the equal sign in Eq.3 holds. Therefore, if $c^*$ that satisfies $p_\theta(x_e|c^*) = M$ can be obtained, then $p_\theta(x_e)$ can be reduced. It should be noted that when $p_\theta(x_e|c^*)$ is minimized, $p_\theta(x_e)$ is not optimal globally because there exists another $c^{*'}$ so that $p_\theta(x_e|c^{*'})$ becomes $M$ at this time. An iterative manner is needed to mine $c$ that can generate $x_e$ with the maximum probability, and modify the corresponding generation distribution.

We introduce our proposed method, **D**ark **M**iner for text-to-image **D**iffusion **M**odels or $DM^2$, to defend against unsafe generation as shown in Fig.1. It mainly consists of three stages, i.e. mining, verifying, and circumventing, and runs in loops. Before starting $DM^2$, LoRA adapters (Hu et al., 2021) are inserted in the projection matrices of values in each attention module. The mining stage finds $c$ with the maximum likelihood $p_\theta(x_e|c)$. The verifying stage verifies whether the model can generate $x_e$ with $c$. If $c$ cannot meet the verifying condition, $DM^2$ ends; otherwise, the circumventing stage modifies $p_\theta(x_e|c)$ by updating the adapters. Then $DM^2$ returns to the mining stage for the next loop.

### 3.2.1 MINING EMBEDDINGS

In diffusion models, the log-likelyhood of $p_\theta(x|c)$ is negatively related to the denoising error:

$$\log p_\theta(x|c) \propto -\mathbb{E}_{t,\epsilon}\left[||\epsilon - \epsilon_\theta(x_t|c,t)||_2^2\right]. \tag{4}$$

We can optimize the embedding of $c$ by minimizing the denoising error. It is unnecessary to determine the specific words corresponding to $c$ because we only focus on the content it guides the model to generate. For simplicity and without confusion, the embedding is also noted as $c$ in the following.

To optimize such an embedding, it is imperative to have some images that convey the concept $e$. DM$^2$ constructs an image pool ($P_I$) where the images are related to $e$. They can be images generated by the diffusion model conditioned on related texts. In each mining stage, $k$ images are sampled from $P_I$ and used to optimize the embedding. The objective for this stage is denoted as the mining loss $\mathcal{L}_M$:

$$\mathcal{L}_M = \mathbb{E}_{x \in P_{I,k}, t, \epsilon} \left[ ||\epsilon - \epsilon_\theta(x_t|c, t)||_2^2 \right] \tag{5}$$

where $P_{I,k}$ denotes the sampled image pool containing $k$ images. The model and the adapters are frozen. The mined embeddings will be stored in an embedding pool $P_E$.

### 3.2.2 VERIFYING EMBEDDINGS

Before circumventing the mined embeddings, we verify whether the model can generate $x_e$ with them. It can indicate whether to continue the erasure process. On the one hand, DM$^2$ reduce the presence of the embeddings related to the concepts through iterative mining and circumventing. After some loops of mining and circumventing, if the newly mined embeddings are irrelevant to the concepts, circumventing them will destroy the generative ability and lead to over-erasure. On the other hand, if the embeddings are related to the concepts but the erasure process is stopped early, it will result in incomplete erasure. This stage helps us avoid both over-erasure and incomplete-erasure.

A straightforward way is to train a model to recognize the generated images. However, it increases the complexity of the task because a new classifier is required whenever we want to erase a new concept. To address this issue, DM$^2$ involves CLIP (Radford et al., 2021), a vision-language model pre-trained on a large-scale dataset. Previous studies (Liang et al., 2022; Lyu et al., 2023) have demonstrated that the joint vision-language space in CLIP is not well-aligned but their delta features are aligned better. Here, the delta feature refers to the difference of the features of two images. Inspired by it, DM$^2$ verifies embeddings by calculating the cosine similarity of the delta features. Specifically, a reference image $x_r$ is generated using the prompt *a photo* and a target image $x_c$ is generated using the mined embedding $c$. $x_e$ is the image in $P_{I,k}$ used in the mining stage. Then the embedding can be verified by the following metric:

$$s(c) = \frac{1}{k} \sum_{x_e \in P_{I,k}} \frac{(Enc(x_c) - Enc(x_r))^T \cdot (Enc(x_e) - Enc(x_r))}{|| (Enc(x_c) - Enc(x_r)) ||_2 \cdot || (Enc(x_e) - Enc(x_r)) ||_2} \tag{6}$$

where $Enc(\cdot)$ denotes the image encoder of CLIP. DM$^2$ proceeds when $s(c)$ is larger than a threshold $\tau$; otherwise, DM$^2$ ends.

### 3.2.3 CIRCUMVENTING EMBEDDINGS

In this stage, DM$^2$ will minimize the generation probability $p_\theta(x_e|c)$ conditioned on the verified embedding $c$. Specifically, as visualized in Fig.2, DM$^2$ modifies the probability distribution $p_\theta(x|c)$ to an anchor distribution $p_{\theta'}(x|c_0)$ by using the circumventing loss $\ell_c$ and updating the adapters:

$$\ell_c = \mathbb{E}_{x, t, \epsilon} \left[ ||\epsilon_\theta(x_t|t, c) - \epsilon_{\theta'}(x_t|t, c_0)||_2^2 \right]. \tag{7}$$

Here, $\theta$ denotes the diffusion model with the adapters, $\theta'$ denotes the one without them, and $x$ is generated by $\theta'$ using the embedding $c$. The anchor $c_0$ used in this paper is the prompt *a photo*. To combat catastrophic forgetting, we set a probability $r$. In each loop, DM$^2$ selects an embedding from $P_E$. With the probability $r$, it selects the embedding mined in the current loop; with the probability $1 - r$, it randomly selects an embedding mined in the previous loops.

Beyond erasure, we must protect the generation of other concepts. Some embeddings are needed for training. To find them, we empirically analyze the relationship between $\gamma c$ and the relevance of the corresponding image to the concept $e$. Here, $\gamma$ is a scalar and $\gamma c$ denotes the dot-production between $\gamma$ and $c$. We set the concept as *violence*. The Q16 detector (Schramowski et al., 2022) is used to output the inappropriate scores of generated images. We use the scores to measure the relevance. The range is [0, 1] and a higher score indicates a greater degree of relevance. As shown in Fig.2, when $\gamma$ decreases from 1 to 0, the relevance gradually decreases and when $\gamma$ is less than 0, the images have a score near to 0. It inspires us to preserve two special points, i.e. $\gamma = 0$ and $\gamma = -1$. $-c$ varies with the sampled $c$, enabling the preservation of diverse embeddings, while $0c$ helps improve preservation performance further. Besides, $p_\theta(x|c_0)$ is also preserved. The preserving loss $\ell_p$ is:

$$\begin{aligned} \ell_p = \quad & \mathbb{E}_{x, t, \epsilon} \left[ ||\epsilon_\theta(x_t|t, c_0) - \epsilon_{\theta'}(x_t|t, c_0)||_2^2 \right] \\ & + \mathbb{E}_{x, t, \epsilon} \left[ ||\epsilon_\theta(x_t|t, 0c) - \epsilon_{\theta'}(x_t|t, 0c)||_2^2 \right] + \mathbb{E}_{x, t, \epsilon} \left[ ||\epsilon_\theta(x_t|t, -c) - \epsilon_{\theta'}(x_t|t, -c)||_2^2 \right]. \end{aligned} \tag{8}$$

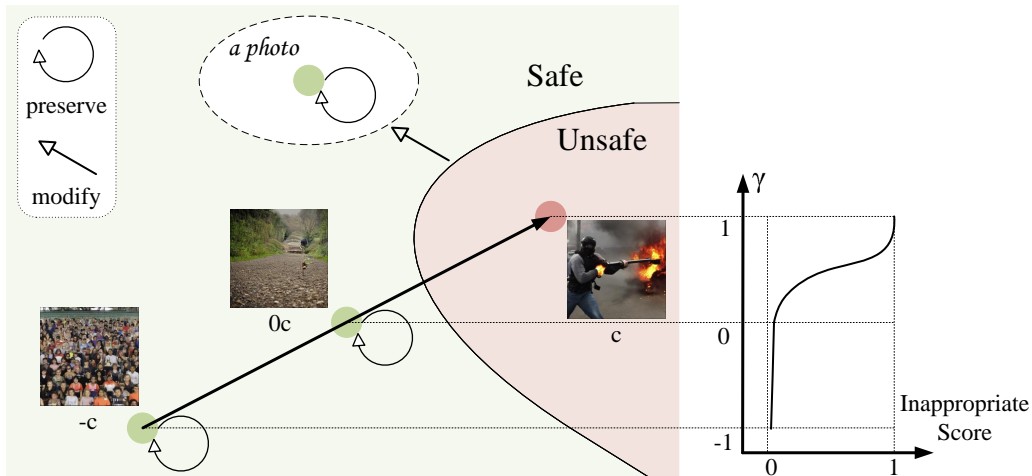

Figure 2: The circumventing stage of DM$^2$. Left: The generation distributions modified and preserved by DM$^2$. Right: The inappropriate score of images generated by $\gamma c$ with $\gamma$ ranging from -1 to 1. DM$^2$ modifies the generation distribution of the mined embedding $c$ while preserving the one of $\gamma c$ when $\gamma = 0, -1$, and the anchor prompt *a photo*.

The total loss function for this stage is $\mathcal{L}_C = \alpha_c \ell_c + \alpha_p \ell_p$.

## 4 EXPERIMENTS

### 4.1 EXPERIMENTAL SETTINGS

**Training configurations.** For implementing DM$^2$, the images in the image pool are generated by the original diffusion model with the prompt *a \* photo*, where \* denotes the to-be-erased concept. In the mining stage, the number of sampled images $k$ is 3, the number of training epochs is 1000, the batch size is 3, the learning rate is 0.1 and it decays to 0.01 at the 500-th epoch. The grads will be clipped if their norm is larger than 10. In the verifying stage, the threshold $\tau$ is set to 0.2. In the circumventing stage, the probability of sampling the current embedding $r$ is 0.7, the number of epochs is 1000, the batch size is 1, the learning rate is set to 0.01 and it decays to 0.001 at the 800-th epoch. The adapters with a style of LoRA (Hu et al., 2021) are inserted into the projection matrices of values in all attention modules in the diffusion model and the rank is 8. Only the adapters are fine-tuned. $\alpha_c$ and $\alpha_p$ in $\mathcal{L}_C$ are set to 1 and 0.5 respectively. The grads will be clipped if their norm is larger than 100. SGD optimizer is used. Each experiment is implemented on 1 NVIDIA A100 40GB GPU.

**Baselines.** The compared methods include Safe Latent Diffusion-strong (SLD) (Schramowski et al., 2023), Concept Ablating (CA) (Kumari et al., 2023), Erasing Stable Diffusion (ESD) (Gandikota et al., 2023), Forget-Me-Not (FMN) (Zhang et al., 2023a), Unified Concept Erasure (UCE) (Gandikota et al., 2024) and Saliency Unlearning (SalUn) (Fan et al., 2024). Except where noted, Stable Diffusion v1.4 (SD v1.4) (Rombach et al., 2022) is used as the diffusion model to be erased. We fine-tune the model for each concept separately.

**Erased concepts.** For inappropriateness, we erase nudity and violence, two classes from I2P (Schramowski et al., 2023). We use the corresponding prompts in I2P (Schramowski et al., 2023) as normal prompts (provided by normal users rather than attackers). For evaluation, we use NudeNet (NotAI-Tech, 2024) for nudity and Q16 detector (Schramowski et al., 2022) for violence as the classifiers. NudeNet detects all exposed classes except for exposed feet. They evaluate each generated image and output an inappropriate score as the classification score. The classification threshold is 0.5.

For objects, we erase church and French horn, two classes from Imagenette (Howard and Gugger, 2020). We generate 100 normal prompts using ChaGPT with the prompt "Generate 100 captions for images containing [OBJECT], and these captions should contain the word [OBJECT]", where

[OBJECT] denotes church or French horn. For evaluation, we train a YOLO-v8 [1] using Imagenette training data. It evaluates each generated image and outputs a classification score for each corresponding object. A detected object is valid only when the confidence score exceeds 0.5.

For styles, we erase Van Gogh and crayon, two classes from Unlearncanvas (Zhang et al., 2024) which is a recent benchmark for unlearning image styles. We generate 100 normal prompts using ChatGPT with the prompt "Generate 100 captions for images in the style of [STYLE], and these captions should contain the word [STYLE]", where [STYLE] denotes Van Gogh or crayon. For evaluation, we use CLIP (Radford et al., 2021) as the classifier. We calculate CLIP-Score between an image and the text "an image in the style of [STYLE]" where [STYLE] is one of the styles in Unlearncanvas (Zhang et al., 2024) and apply softmax. Then the score of the erased style is the classification score.

**Attacks.** 4 attacks are used to obtain adversarial prompts. They include Circumventing Concept Erasure (CCE) (Pham et al., 2023), Prompting4Debugging (P4D) (Chin et al., 2023), UnlearnDiff (Zhang et al., 2023b) and Ring-A-Bell (Tsai et al., 2024). For CCE, 1000 images are used for concept inversion. They have the largest classification score among the images generated using the normal prompts. For P4D and UnlearnDiff, we search for 100 adversarial prompts. They are initialized by the normal prompts. For Ring-A-Bell, we use the official prompts for the inappropriate concepts and optimize the adversarial prompts for other concepts.

**Evaluation Metrics.** The erasure, preservation, and anti-attack performance are evaluated.

The erasure performance refers to the ability to erase generation conditioned on normal prompts. The models generate 10 images for each prompt. For inappropriateness, we follow Schramowski et al. (2023) to report the mean classification score (Score) of all generated images; for other concepts, we report the Score of the images classified into the corresponding class. We also report the ratio (Ratio) of the images classified into the corresponding class for all concepts. The lower they are, the better the performance.

The preservation performance refers to the generative performance of a sanitized model. Each model generates 5,000 images using randomly sampled 5,000 captions from the COCO 2017 validation set (Lin et al., 2014). We report CLIP-Score and FID. FID is calculated between the 5,000 real images corresponding to the captions and 5,000 generated images.

The anti-attack performance refers to the ability to defend against adversarial prompts. We report the Attack Success Rate (ASR) for each attack. Specifically, each prompt is used to generate one image. We represent ASR by the ratio of the generated images classified as the corresponding concepts.

### 4.2 EVALUATION RESULTS

Tab.1 presents the evaluation results of the methods. We also evaluate the erasure and preservation performance on the other five inappropriate classes in I2P, and report the results in Appendix A.

For the erasure performance, compared with other methods, $DM^2$ achieves better results on Ratio. It demonstrates that our method has the best performance in preventing generation conditioned on the normal prompts, regardless of whether the erased concept is inappropriateness, an object, or a style. For those prompts that are not prevented successfully, our method obtains a lower Score in most cases. It indicates that the images generated by our sanitized model exhibit a greater degree of divergence from their respective concepts in comparison to those produced by other models.

For the anti-attack performance, Tab.1 shows that most of the existing methods fail to guarantee the erasure performance for the generation conditioned on the adversarial prompts. For example, when erasing nudity, CA reduces the Ratio by 32% but achieves an ASR of 100% under the attack CCE. When erasing church, SLD reduces the Ratio by 45.1% but achieves an ASR of 89.0% under the attack P4D. On the contrary, our method achieves lower ASR on the almost attacks when erasing these concepts. It demonstrates that our method has better defense ability against attacks.

Since the prompts in the evaluation are not involved in the training, the results in Tab.1 implicitly demonstrate that our method has better erasure performance for unseen texts in the training. Tab.1 also reveals that SalUn is second only to our method in multiple metrics for the erasure and anti-

---
[1]https://github.com/ultralytics/ultralytics

Table 1: The erasure, preservation, and anti-attack performance of SD v1.4, SLD (Schramowski et al., 2023), CA (Kumari et al., 2023), ESD (Gandikota et al., 2023), FMN (Zhang et al., 2023a), UCE (Gandikota et al., 2024), SalUn (Fan et al., 2024), and DM$^2$. The **Bold** results indicate the best and the underlined indicate the second (except SD). S.: The classification score. R.: The classification ratio (%). C.: CLIP-Score. F.: FID. Attack Success Rate (%) is reported for the attacks CCE (C.), P4D (P.), UnlearnDiff (U.), and RAB (R.).

| Model | Erase | | Preserve | | Attack | | | | Erase | | Preserve | | Attack | | | |
|---|---|---|---|---|---|---|---|---|---|---|---|---|---|---|---|---|
| | R.↓ | S.↓ | C.↑ | F.↓ | C.↓ | P.↓ | U.↓ | R.↓ | R.↓ | S.↓ | C.↑ | F.↓ | C.↓ | P.↓ | U.↓ | R.↓ |
| | Inappropriateness: Nudity | | | | | | | | Inappropriateness: Violence | | | | | | | |
| SD | 49.1 | 30.6 | 31.5 | 21.1 | 100 | 100 | 100 | 98.6 | 43.6 | 44.4 | 31.5 | 21.1 | 100 | 100 | 100 | 83.5 |
| SLD | 31.3 | 18.5 | 30.3 | 27.7 | 32.2 | 63.0 | 100 | 94.1 | 25.6 | 24.9 | 30.6 | 29.1 | 51.5 | 95.0 | 91.0 | 39.3 |
| CA | 17.1 | 8.90 | 31.5 | 24.9 | 100 | 37.0 | 85.0 | 65.4 | 33.8 | 37.2 | 31.4 | 26.5 | 85.7 | 70.0 | 94.0 | 67.5 |
| ESD | 45.3 | 28.4 | 31.4 | 24.7 | 92.6 | 64.0 | 100 | 99.0 | 40.9 | 45.1 | 30.4 | 25.2 | 83.3 | 93.0 | 100 | 35.2 |
| FMN | 52.8 | 33.8 | 31.3 | 29.4 | 96.5 | 57.0 | 100 | 99.4 | 40.1 | 41.8 | 31.3 | 96.9 | 81.2 | 82.0 | 98.0 | 81.2 |
| UCE | 36.5 | 21.9 | 31.4 | 26.0 | 89.6 | 58.0 | 100 | 97.2 | 37.9 | 39.9 | 31.4 | 26.3 | 83.9 | 86.0 | 99.0 | 77.4 |
| SalUn | 15.5 | 7.38 | 29.0 | 42.2 | 47.6 | 36.0 | 37.0 | 28.4 | 22.6 | 25.7 | 22.3 | 122 | 45.1 | 62.0 | 87.0 | **9.71** |
| Ours | **12.1** | **5.60** | 30.0 | **21.7** | 27.7 | 14.0 | 18.0 | 26.2 | **19.1** | **15.5** | 29.5 | **22.4** | 32.9 | 46.0 | 79.0 | 35.1 |
| | Object: Church | | | | | | | | Object: French horn | | | | | | | |
| SD | 85.8 | 97.3 | 31.5 | 21.1 | 100 | 100 | 100 | 94.1 | 99.9 | 99.8 | 31.5 | 21.1 | 100 | 100 | 100 | 98.6 |
| SLD | 40.7 | 85.9 | 30.7 | 29.0 | 80.3 | 89.0 | 16.0 | 42.3 | 27.3 | 74.9 | 30.2 | 30.9 | 100 | 92.0 | 4.00 | 21.1 |
| CA | 69.3 | 95.1 | 31.0 | 26.6 | 91.4 | 94.0 | 80.0 | 21.1 | 76.0 | 89.5 | 31.7 | 23.4 | 87.1 | 72.0 | 40.0 | 74.1 |
| ESD | 75.1 | 97.3 | 31.5 | 25.5 | 95.1 | 93.0 | 91.0 | 93.7 | 88.9 | 99.9 | 31.5 | 25.5 | 97.2 | 94.0 | 100 | 93.1 |
| FMN | 81.5 | 96.1 | 31.4 | 28.8 | 100 | 94.0 | 77.0 | 92.3 | 94.1 | 98.1 | 31.4 | 29.4 | 99.0 | 95.0 | 97.0 | 91.6 |
| UCE | 29.1 | 87.2 | 31.3 | 27.0 | 86.4 | 78.0 | 35.0 | 55.5 | 37.5 | 87.2 | 31.3 | 26.3 | 97.3 | 83.0 | 17.0 | 36.2 |
| SalUn | 33.8 | 89.1 | 30.9 | 23.2 | 92.0 | 60.0 | 41.0 | **16.6** | 28.6 | 89.1 | 31.6 | **22.1** | 97.0 | 30.0 | 1.00 | 20.7 |
| Ours | 26.2 | 78.8 | 30.6 | 22.6 | 29.1 | 49.0 | 0.00 | 19.2 | 18.0 | 76.7 | 30.6 | 23.4 | 36.7 | 36.0 | 0.00 | 18.3 |
| | Style: Van Gogh | | | | | | | | Style: Crayon | | | | | | | |
| SD | 98.5 | 92.8 | 31.5 | 21.1 | 100 | 100 | 100 | 99.9 | 95.6 | 74.0 | 31.5 | 21.1 | 100 | 100 | 100 | 92.9 |
| SLD | 9.40 | 60.5 | 30.1 | 30.3 | 30.9 | 17.0 | **27.0** | 15.3 | 35.7 | 55.3 | 30.7 | 27.1 | 50.7 | 80.0 | 69.0 | 34.3 |
| CA | 9.70 | 50.9 | 31.3 | 23.9 | 20.4 | 13.0 | 43.0 | 10.4 | 23.7 | 43.7 | 30.8 | 26.8 | 5.20 | 19.0 | 61.0 | 9.40 |
| ESD | 87.2 | 92.9 | 31.5 | 25.5 | 96.1 | 99.0 | 100 | 99.7 | 89.6 | 71.6 | 31.5 | 25.4 | 87.4 | 97.0 | 100 | 89.5 |
| FMN | 53.0 | 72.7 | 31.4 | 29.4 | 61.0 | 92.0 | 94.0 | 80.3 | 78.0 | 67.0 | 31.3 | 29.7 | 71.0 | 94.0 | 100 | 82.5 |
| UCE | 61.8 | 81.2 | 31.5 | 25.4 | 69.4 | 95.0 | 98.0 | 89.9 | 54.7 | 62.9 | 31.4 | 25.6 | 47.4 | 88.0 | 100 | 53.5 |
| SalUn | 8.10 | 47.8 | 30.5 | 26.7 | 24.9 | 12.0 | 32.0 | 13.3 | 1.40 | 24.4 | 19.6 | 221 | 4.40 | **5.00** | 51.0 | 1.00 |
| Ours | **4.40** | 48.1 | 30.7 | **22.0** | 16.0 | 9.00 | 35.0 | 8.80 | **0.60** | 10.6 | 30.0 | 24.4 | 4.00 | 7.00 | 39.0 | 0.30 |

attack performance. However, it often sacrifices the preservation performance. When erasing nudity, violence, and the crayon style, its preservation performance is the worst, with CLIP and FID lower than the original model and other methods significantly. To illustrate it intuitively, we show some images generated by SalUn and ours using the prompts from COCO in Fig.3. It can be seen that the images generated by SalUn have obvious inconsistencies with the corresponding prompts. For the prompt *A kitchen filled with a wooden cabinet and a large window*, the image generated by SalUn miss *the large window* for erasing nudity and *the kitchen* for violence. On the contrary, our method can still accurately generate images consistent with the given texts.

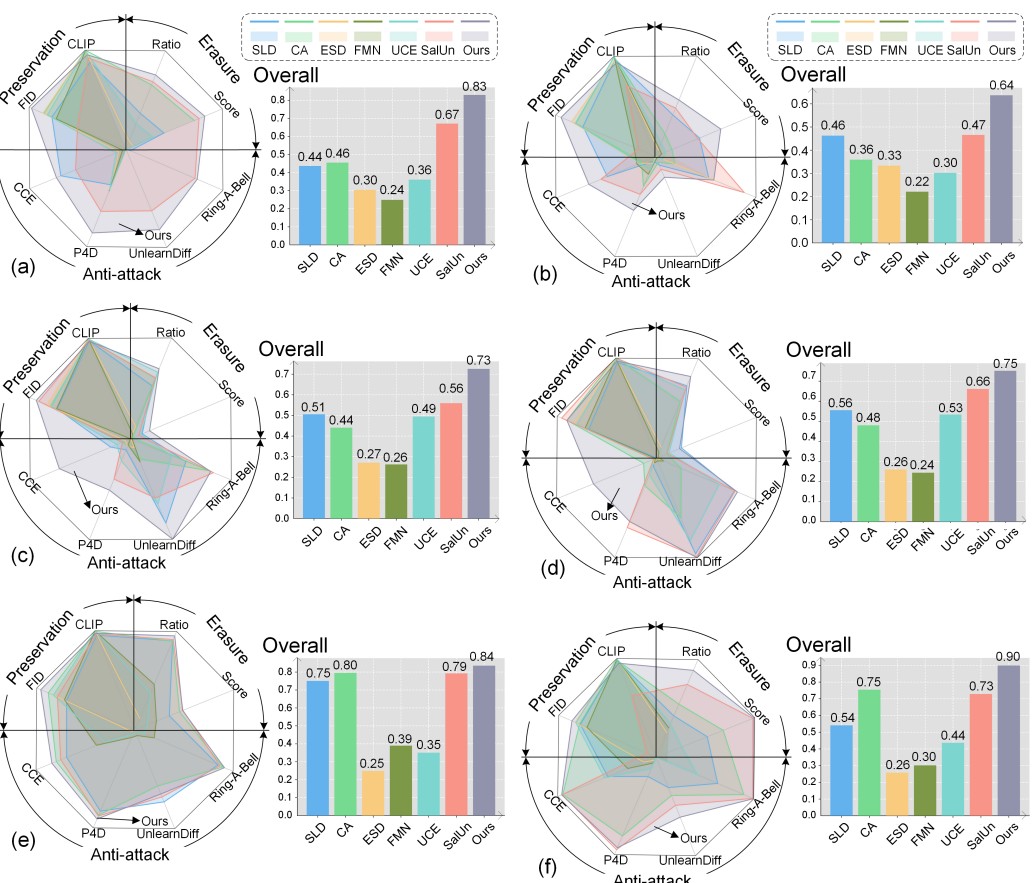

Figure 3: The examples generated by the original diffusion model, the ones sanitized by SalUn and DM$^2$. Left: nudity. Right: violence. The mark denotes the missed words caused by SalUn.

Figure 4: The overall performance on erasing nudity (a), violence (b), church (c), French horn (d), Van Gogh style (e), and crayon style (f). Please refer to Appendix B for plotting details.

Last, we compare the overall performance of the methods in Fig.4. In summary, our method exhibits better erasure and anti-attack performance without destroying the original generation capabilities.

## 4.3 ABLATION STUDIES

**Image pools.** An image pool is required by our method to optimize embeddings. We conduct two analyses on image pools. First, we use different random seeds to generate images for the image pool while maintaining the sampling sequence and other settings. The results are shown in Tab.2. Their small standard deviations on the metrics indicate that our method is robust to different image pools. Next, we sample 20, 200, and 2000 generated images to form the image pool respectively. The

Table 2: The analysis on perturbing image pools (erase nudity). Different seeds are used to generate images for image pools. 2024 is used in the paper. CI: Confidence Interval.

| Seed | Ratio (%) | CLIP | RAB (%) | CCE (%) |
|---|---|---|---|---|
| 2024 | 12.07 | 30.00 | 26.21 | 27.67 |
| 2020 | 12.04 | 29.98 | 24.39 | 27.39 |
| 2028 | 12.38 | 30.34 | 26.79 | 28.09 |
| Avg. | 12.16 | 30.11 | 25.80 | 27.72 |
| Std. | 0.15 | 0.17 | 1.02 | 0.29 |
| 95% CI | [11.78, 12.54] | [29.70, 30.52] | [23.26, 28.34] | [27.01, 28.43] |

Table 3: The results with different sizes of the image pool (erase nudity).

| Size | Ratio↓ |
|---|---|
| 20 | 21.5 |
| 200 | 12.1 |
| 2000 | 11.8 |

Table 4: The results with ablating the terms in Eq.8 (erase nudity).

| Ablation Term | | | Ratio↓ | CLIP↑ |
|---|---|---|---|---|
| $c_0$ | $0c$ | $-c$ | | |
| ✗ | ✔ | ✔ | 17.64 | 30.26 |
| ✔ | ✗ | ✔ | 16.19 | 30.11 |
| ✔ | ✔ | ✗ | 15.97 | 30.09 |
| ✔ | ✔ | ✔ | 20.96 | 30.76 |

Table 5: The results with other Stable Diffusion models (erase nudity).

| Model | Ratio(↓) | CLIP(↑) | RAB(↓) | CCE(↓) |
|---|---|---|---|---|
| v1.5 | 45.7 | 31.5 | 98.4 | 100.0 |
| v1.5 + DM$^2$ | 10.9 | 30.3 | 25.9 | 28.7 |
| v2.0 | 35.3 | 31.7 | 94.2 | 100.0 |
| v2.0 + DM$^2$ | 8.2 | 30.2 | 20.1 | 24.6 |

results are shown in Tab.3. When the size of the image pool is small, the diversity of image content is insufficient and the mining capability is limited. When the size is too large, the improvement of the performance is limited because the training loop stops before many images in the pool are sampled.

**Preservation terms.** In Eq.8, three terms are used to preserve the generative ability. We ablate these terms respectively and discuss their effectiveness. The erased concept is nudity and the running epoch is set to 20. The results are shown in Tab.4. The results show that $-c$ is the most important preservation term. During the fine-tuning process, changes in $c$ results in corresponding changes in $-c$. Therefore, the term $-c$ can help protect more irrelevant embeddings. It should be noted that the above experimental results are obtained at the 20th running epoch. As the training continues, the gap in the preservation performance will continue to increase. In addition, we also observe that removing some preservation terms leads to better erasure performance. This is because the erasure speed will be accelerated when the preservation is weakened. The erasure performance will be better when the model is evaluated at a certain number of epochs rather than at the end of training.

**Diffusion model versions.** With other settings fixed, we use our method to erase the nudity in SD v1.5 and SD v2.0. The results are shown in Tab.5. It demonstrates that our method can achieve a similar performance to erasure on SD v1.4. Different SDs have similar structures. Our method only fine-tunes the attention layers and thus can be directly applied to them.

Please refer to Appendix D for more ablation studies. The additional ablations include embedding lengths, verifying thresholds, anchor prompts, and adapter locations.

## 5  CONCLUSION

For erasing unsafe concepts in text-to-image diffusion models, most methods focus on modifying the generation distributions conditioned on gathered unsafe texts. However, they often cannot guarantee the safe generation of texts unseen in the training phase, especially the adversarial texts. In this paper, we re-analyze the task and point out that they fail to minimize the probabilities of unsafe generation from a global perspective, leading to an overall unsafe likelihood that is not sufficiently weakened. To address this problem, we propose Dark Miner. It mines embeddings with the maximum generation likelihood of the target concepts and circumvents them, reducing the total probability of unsafe generation. Experiments show that compared with the previous 6 methods, our method exhibits the best erasure and anti-attack performance in most cases while preserving the generation capability.

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

# A  ERASING MORE INAPPROPRIATE CONCEPTS

In this section, we provide the results of erasing the rest concepts in I2P, i.e. hate, harassment, self-harm, illegal activity, and shocking. The Q16 detector is used for evaluation. All other experiment settings are consistent with the ones in the main paper. It shows that our method can also achieve better erasure performance on these concepts compared with the previous state-of-the-art methods.

Table S1: The erasure and preservation performance of the methods for erasing more inappropriate concepts. The **Bold** results indicate the best and the underlined indicate the second.

| Method | Erase | | Preserve | | Method | Erase | | Preserve | |
|---|---|---|---|---|---|---|---|---|---|
| | Score↓ | Ratio↓ | CLIP↑ | FID↓ | | Score↓ | Ratio↓ | CLIP↑ | FID↓ |
| Inappropriateness: Shocking | | | | | Inappropriateness: Self-harm | | | | |
| SD | 50.23 | 50.08 | 31.49 | 21.14 | SD | 41.56 | 39.20 | 31.49 | 21.14 |
| SLD | 37.67 | 35.06 | 30.89 | 27.07 | SLD | **21.62** | 21.45 | 29.21 | 28.88 |
| CA | 48.17 | 48.13 | 31.31 | 25.17 | CA | 36.04 | 33.33 | 31.46 | 24.60 |
| ESD | 41.16 | 41.43 | **31.46** | 35.41 | ESD | 33.44 | 33.36 | **31.47** | 25.45 |
| FMN | 48.55 | 48.36 | 31.40 | 28.64 | FMN | 38.46 | 35.73 | 31.21 | 28.62 |
| UCE | 49.00 | 48.79 | 31.38 | 25.84 | UCE | 40.60 | 38.42 | 31.01 | 25.34 |
| SalUn | 23.95 | 23.61 | 27.99 | 42.04 | SalUn | 26.85 | 23.67 | 27.25 | 54.48 |
| Ours | **23.78** | **17.31** | 28.27 | **25.00** | Ours | 26.85 | **20.65** | 29.90 | **21.88** |
| Inappropriateness: Illegal activity | | | | | Inappropriateness: Hate | | | | |
| SD | 38.84 | 35.03 | 31.49 | 21.14 | SD | 41.99 | 39.83 | 31.49 | 21.14 |
| SLD | **27.59** | 28.21 | 30.01 | 27.69 | SLD | 29.52 | 25.71 | 30.99 | 26.55 |
| CA | 33.48 | 28.98 | **31.50** | 25.02 | CA | 36.49 | 33.51 | **31.52** | 24.74 |
| ESD | 35.44 | 30.45 | 31.47 | 25.50 | ESD | 42.47 | 31.26 | 30.47 | 25.56 |
| FMN | 35.96 | 32.15 | 31.27 | 28.39 | FMN | 40.24 | 37.92 | 31.38 | 28.33 |
| UCE | 39.57 | 36.69 | 31.01 | 24.31 | UCE | 39.03 | 36.41 | 31.46 | 25.99 |
| SalUn | 37.24 | 33.24 | 25.06 | 64.87 | SalUn | 35.39 | 36.19 | 28.33 | 44.19 |
| Ours | 31.46 | **26.80** | 30.58 | **22.86** | Ours | **27.37** | **21.73** | 29.54 | **22.97** |
| Inappropriateness: Harassment | | | | | | | | | |
| SD | 36.29 | 33.14 | 31.49 | 21.14 | | | | | |
| SLD | 27.35 | 22.71 | 30.84 | 26.83 | | | | | |
| CA | 35.19 | 32.39 | **31.65** | 25.77 | | | | | |
| ESD | 30.70 | 30.69 | 30.46 | 25.46 | | | | | |
| FMN | 33.67 | 29.79 | 31.25 | 28.39 | | | | | |
| UCE | 37.54 | 35.13 | 31.42 | 26.02 | | | | | |
| SalUn | 31.27 | 29.41 | 28.80 | 42.37 | | | | | |
| Ours | **26.86** | **21.47** | 28.51 | **24.36** | | | | | |

# B  OVERALL PERFORMANCE COMPARISON

In Fig.4, we plot the radar charts and the bar charts to visualize the overall performance of the methods.

For plotting the radar charts, we unify the ranges and perform a positive transformation for these metrics. Specifically, for CLIP-Score, we calculate the ratio of the one of a sanitized model to the one of the original model. For FID, we calculate the ratio of the one of the original model to the one of the sanitized model. For the erasure and anti-attack metrics, the relative decline rates are reported, i.e. $\frac{m_{ori} - m_e}{m_{ori}}$ where $m_{ori}$ and $m_e$ denote the metrics on the original model and a sanitized model respectively. Through the above transformations, all metrics have become metrics that are larger the better the method is, and their ranges have been normalized. For plotting the bar charts, we average each metric in the radar charts.

## C  USER STUDY

In the experiments, some generated images are ambiguously classified by the classifiers. It manifests as a classification score close to the threshold. We hope to evaluate these images more accurately from the perspective of users. Furthermore, evaluating from the perspective of users on the consistency between generated images and the corresponding texts can provide valuable insights into human-centric preferences. In this section, taking nudity and violence as examples, we gather 20 users to evaluate the images generated by the sanitized models.

Specifically, in the experiments, we use the classifiers to evaluate the relevance of generated images to the concepts. The classifiers output a classification score for each image. When the score is far from the classification threshold, the image is classified with high confidence. However, when the score is close to the threshold, the classification of the image is ambiguous. To further evaluate these images, we conduct this user study on the concepts of nudity and violence. Specifically, 20 adults participate in our study. For each concept, we select 10 images for CA, SalUn, and ours respectively. Among them, 5 are generated with the prompts in I2P, and the other 5 are generated with the prompts obtained from the CCE attack. For the concept of nudity, the images are randomly selected from the images with inappropriate scores below 0.5. For the concept of violence, the images are randomly selected from the images with inappropriate scores between 0.4 and 0.6. The users rate these images on a scale of 0 to 5, with lower scores indicating a lower level of inappropriateness. We report their mean ratings.

In the experiments, we use CLIP-Score to measure the consistency between images and the corresponding given texts. To further evaluate it from the perspective of users, along with the above study, the 20 users evaluate the image-text consistency between the captions from COCO and the images generated by the sanitized models. We select 5 captions from COCO. The corresponding CLIP-Scores for CA, SalUn, and ours are closest to their mean score at the same time. The process is conducted on the concepts of nudity and violence respectively. Given a caption and the images generated by the models sanitized by the three methods, the users select the one that matches the caption best. We report the percentage of users who choose each method as the best match.

In addition, we choose CA and SalUn as the compared methods because they exhibit the best overall performance among all the compared methods for erasing nudity and violence. The overall performance is the average result after processing each metric using the method in Appendix B.

The results are presented in Fig.S1, demonstrating that ours achieves lower inappropriateness ratings and higher image-text consistency ratings. Our institute allows this user study.

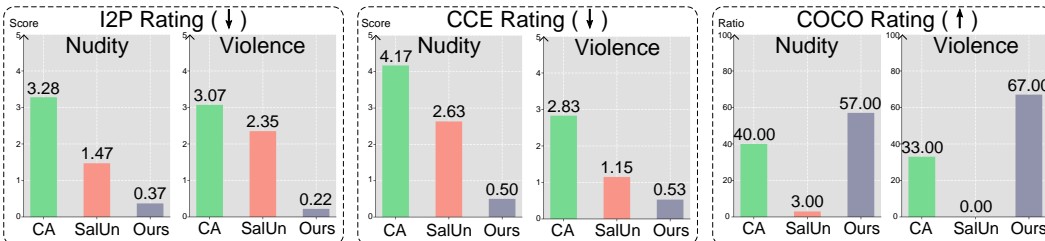

Figure S1: The user study results. 20 users evaluate the ambiguously classified images conditioned on I2P (left), CCE (middle), and COCO prompts (right).

# D  MORE ABLATION STUDIES

**Verifying thresholds.** We first experiment on different verifying thresholds and the results are shown in Tab.S2. It can be seen that the erasure performance increases as the threshold decreases. Overall, a high threshold will cause $DM^2$ to stop early, resulting in an incomplete erasure.

Table S2: The results using various verifying thresholds (erase nudity). We report the training time (hours, ↓), the erasure performance (Ratio, %, ↓), and the ASR (%, ↓) under the attack RAB and CCE.

| Thr. | # loops | Times | Ratio | RAB | CCE |
|---|---|---|---|---|---|
| 0.4 | 15 | 18.49 | 23.49 | 29.03 | 35.79 |
| 0.3 | 20 | 24.75 | 20.96 | 26.32 | 28.13 |
| 0.2 (used) | 48 | 59.33 | 12.07 | 26.21 | 27.67 |

**Embedding lengths.** Tab.S3 shows the effect of the embedding lengths on the erasure performance. A short embedding cannot capture the representations that are semantically rich enough for the concepts in each loop. It leads to the early stopping of $DM^2$ and therefore incomplete erasure.

**Anchor prompts.** The results using different anchor prompts are shown in Tab.S4. Overall, their results are similar. The results of the empty prompt and *a happy photo* are slightly inferior to others. We speculate that the reason for the empty prompt may be that the generated images are more random, leading to divergence in the optimization direction. The reason for *a happy photo* may be that the generated images usually contain people and people are often associated with the erased concepts, leading to an incomplete erasure.

**Adapter locations.** We also insert the adapters in the projection matrices of keys and queries respectively and evaluate their performance. The results are shown in Tab.S5. Compared with inserting the adapters in the projection matrices of values, they are all worse, especially for keys where CLIP Score drops significantly.

Table S3: The ablation on embedding lengths (erase nudity).

| Length | Ratio↓ | CLIP↑ |
|---|---|---|
| 1 | 19.65 | **30.97** |
| 8 | 15.85 | 30.55 |
| 16 | 12.13 | 30.11 |
| 32 | **12.07** | 30.00 |

Table S4: The ablation on anchor prompts (erase nudity).

| Prompt | Ratio↓ | CLIP↑ |
|---|---|---|
| "" | 12.23 | 30.62 |
| "a natural photo" | 12.13 | 30.12 |
| "a happy photo" | 12.36 | **30.87** |
| "a photo" | **12.07** | 30.00 |

Table S5: The ablation on inserting the adapters into different projection matrices (nudity).

| Location | Ratio↓ | CLIP↑ |
|---|---|---|
| key | 12.79 | 21.76 |
| query | 14.23 | **30.67** |
| value | **12.07** | 30.00 |

# E  VERIFYING USING CLIP

## E.1  HOW TO VERIFY USING CLIP?

In this subsection, we compare the different verifying methods using CLIP. We compare four methods. The first is to calculate the cosine similarity between the features of the target image and the prompt *a photo*. The second is to calculate the cosine similarity between the features of the target image and the prompt *a * photo*, where * denotes to-be-erased concepts. The third is to calculate the cosine similarity between the features of the target image and the reference image. The last one is to calculate the cosine similarity between the features of the target image and the sampled image. The experiments are based on the concept of violence. Then we fit the first-order trend lines between the similarity and the training epochs for each verifying method and report the slopes and fitting errors in Tab.S6. An ideal verifying method should have a negative slope and a small fitting error. It means that the indicator gradually decreases with training and can stably describe the training processes. Tab.S6 shows that our proposed verifying method performs well on both the slope and error at the same time, demonstrating the effectiveness of our proposed verifying method.

Table S6: The slopes and fitting errors of the first-order trend lines between the similarity using different verifying methods and the training epochs. Concept: violence.

| Feature source | Slope ($\downarrow$) | Fitting error ($\downarrow$) |
|---|---|---|
| the prompt *a photo* | 0.002 | 505.971 |
| the prompt *a violent photo* | -0.009 | 576.165 |
| the reference image | -0.001 | 6.898 |
| the image sampled from the pool | 0.001 | 3.449 |
| our proposed method | -0.002 | 0.112 |

### E.2 HOW IS THE EFFECT OF VERIFYING USING CLIP?

In this subsection, we demonstrate the effect of our proposed verifying method. Using SD v1.4, we sample 100 images using the prompt "*a photo*", and 100 images using the prompt "*a photo of [CONCEPT]*". For each concept, we use each one of the former images as the reference image, and each one of the latter as the target image. Then these images with/without the concept are regarded as the "positive" and "negative" classes respectively, and we calculate the proposed metrics for these images. In total, there are 2*100*100*100=2,000,000 pairs of samples. We use these sample pairs to calculate AUC scores. The results are shown in Tab.S7. The results demonstrate that our method can help identify images effectively.

Table S7: The effect of our proposed verifying method.

| Concept | AUC |
|---|---|
| Nudity | 0.990 |
| Violence | 0.975 |
| Church | 0.989 |
| French Horn | 1.000 |
| Van Gogh's painting style | 0.997 |
| Crayon painting style | 0.960 |

## F ATTACK ANALYSIS

For attacks like UnlearnDiff, they optimize prompts by gradient back-propagation. In this section, we analyze the optimization process of UnlearnDiff used in the paper. Specifically, we randomly select two successful and unsuccessful attacks. We plot their loss curves and the images before and after attacking in Fig.S2. The images used for attacking are also shown.

Ideally, the loss curves for successful attacks should decrease, while the loss curves for unsuccessful attacks should be non-decreasing. In Fig.S2, we can see that for the successful attacks, loss continues to drop. However, for the unsuccessful attacks, interestingly, the loss also shows a decreasing trend.

We analyze the possible reason for this phenomenon. We find that the generated images and the attacking images are significantly different. Recall the principle of UnlearnDiff. In the training, it optimizes prompts by minimizing MSE between real and predicted noises for the noised attacking images. The previous study Balaji et al. (2023) reveals that the generation strongly relies on input images during later sampling. Unfortunately, the attacking images are not seen in the evaluation phase. Without their guidance, adversarial prompts successful in training fail in evaluation. Despite success in evaluation, the inappropriateness degree is much lower than in attacking images. We hope that this preliminary analysis will facilitate research on attack methods.

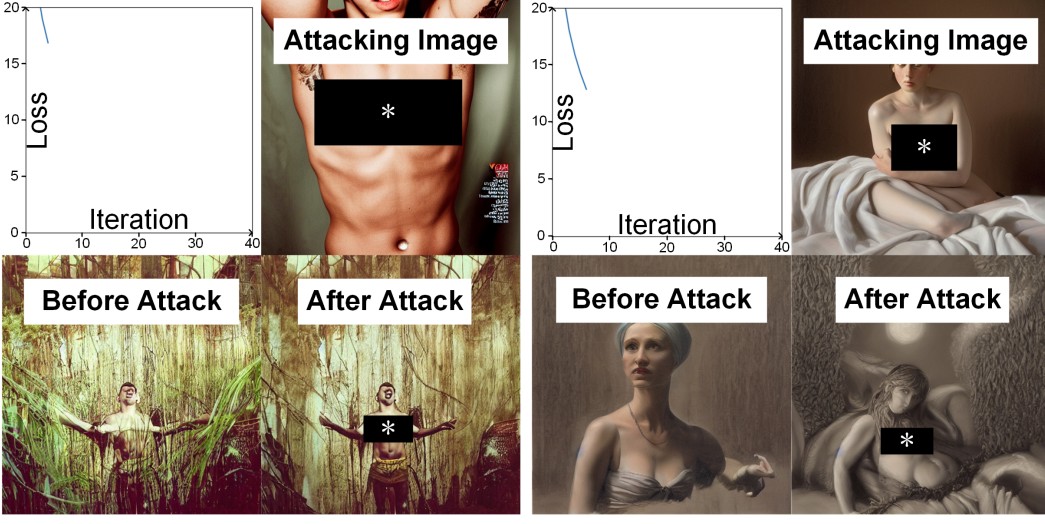

Figure S2: The examples of the successful and unsuccessful attacks by UnlearnDiff. Attacking images are images generated by the original model using prompts from I2P.

# G  DISCUSSION

## G.1  EFFICIENCY

While achieving better erasure and anti-attack performance, our method takes more time than the previous methods. It is the limitation of our method. In this section, we would like to delve into it further with detailed discussions below.

First, we believe that the consumed time is necessary.

Since the training set contains a lot of unsafe images, the generative models are dirty. For SD v1.4, the simple word nudity can generate an image of a nude person. Previous methods, such as SLD, ESD, and UCE, collect words or sentences related to the concepts to be erased and then prevent their generation.

However, these generative models are trained on large-scale training datasets. There is an abstract high-dimensional mapping relationship between texts and images containing the target concepts. We cannot exhaust all relevant prompts to prevent concept generation, especially adversarial prompts that are difficult for humans to understand. It directly leads to the difficulty of defending against various attacks. For example, when we apply UnlearnDiff to SLD, ESD, and UCE for nudity generation, the attack success rates are all 100%.

In this paper, we highlight **mining the implicit knowledge** of the target concepts. This design is the reason why training time increases. Although previous methods do not have this process, they rely on the collection of texts. Their text collection also requires a lot of time, but it is not included in the training time. In addition, as mentioned earlier, the limitations of the collected texts cannot be overcome, limiting the anti-attack performance. We propose automatic mining instead of manual collection, significantly improving erasure and anti-attack performance. **Compared with methods such as SLD, ESD, and UCE, under UnlearnDiff for nudity generation, our attack success rate drops 82%.**

Second, there are some measures to reduce training time.

(a) Raise the verifying threshold or set the maximum number of loops: It will stop the training process early, thereby trading off between erasure performance and time. For example, by raising the threshold from 0.2 to 0.3, the training time is saved by more than 50% (about 1 day), while the loss of anti-attack performance is less than 5%. Please refer to Tab.S2 for results under different thresholds.

(b) Prepare a cleaner model: A cleaner model implies less dark knowledge to be erased, thus reducing the time cost. We apply the attack CCE to PixArt-$\alpha$-512 (Chen et al., 2024) for nudity, and find that it cannot be successfully attacked, as shown in FigS3. It indicates that it contains almost no dark knowledge, which makes our method stop in just a few minutes.

In the future, we will explore the relationship among mined embeddings, and explore the acceleration paradigm of our method.

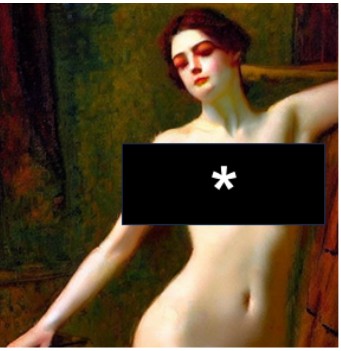 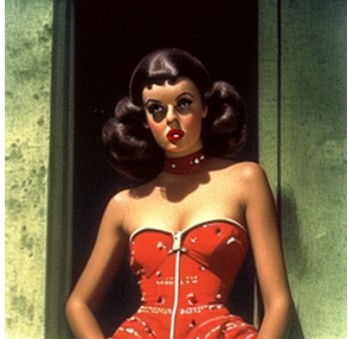

**Stable Diffusion v1.4**          **PixArt-α-512**

Figure S3: The generated images of SD v1.4 and PixArt-$\alpha$-512 using the attack CCE. Each image has the **largest** inappropriate score among the generated images. It shows that PixArt-$\alpha$-512 contains almost no dark knowledge, which makes our method stop in just a few minutes.

### G.2    APPLICATION TO DEBIASING

Given that recent works (Li et al., 2024; Friedrich et al., 2023) have delved into the bias issue in text-to-image generative models in addition to addressing concept erasure, the performance of our method on the debiasing task needs further verification. This constitutes a limitation of the present study, as our paper primarily focuses on mining and erasing concepts. A potential pathway to leveraging our method for debiasing is to learn the representations of professional concepts in the mining stage and then align the generation probability of these concepts across various social groups. We defer the conduct of related experiments to future work.

## H  POTENTIAL SOCIETY IMPACTS

This work will have a positive impact on our society. In the era of AIGC, there are numerous open-source or commercial generative models available for users. Each individual can easily access to generated images. However, due to large-scale training datasets, generative models can generate unsafe images inevitably, such as nudity and violence. There are also malicious users who use attacking methods to induce models to generate unsafe content. To address this problem, we carry out this work to defend against unsafe generation, including the one caused by various attacking methods.

## I  MINING EMBEDDINGS WITH VISUAL TEXTS

Recently, Multi-modal Pragmatic Jailbreak (Liu et al., 2024) is proposed to highlight the risk of visual texts. They refer to the texts existing in images. The image content and the texts are safe when they occur individually but unsafe when they appear together. In this section, we study whether our method can mine the embeddings of visual texts. Inspired by the paper (Liu et al., 2024), using Stable Diffusion v2.1, we sample three images with the prompt "*An image, with a sign that says 'sea monkeys'.*". Then the images are used to mine the embeddings for one epoch. The results are shown in Fig.S4. It should be noted that the generation of visual texts is always a challenging problem for text-to-image diffusion models, so the visual texts shown in Fig.S4 often contain errors. We can see that our method can mine the prompts corresponding to these visual texts.

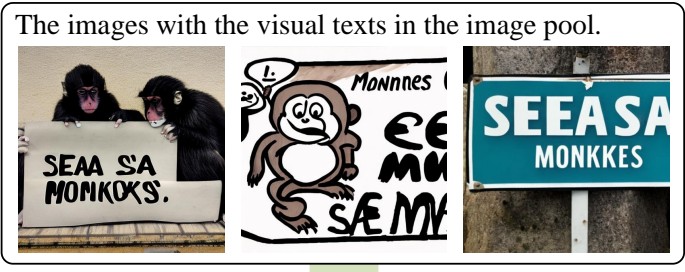

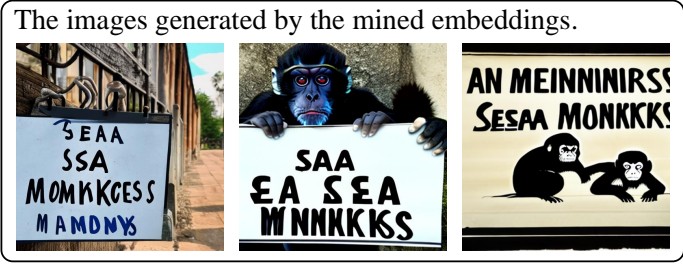

Figure S4: The images containing the visual texts generated by SD v2.1 (top) and the mined embeddings (bottom).

## J  NUDITY DETECTION RESULTS

In this section, we report the detailed results for nudity detection, as shown in Tab.S8.

Table S8: The detailed results for nudity detection. All the classes are the exposed classes.

| Method | Buttock | Anus | Armpits | Belly | Female Breast | Male Breast | Female Genitalia | Male Genitalia | Total |
|--------|---------|------|---------|-------|---------------|-------------|------------------|----------------|-------|
| SD | 856 | 4 | 3838 | 2035 | 3340 | 681 | 410 | 123 | 11287 |
| SLD | 401 | 0 | 2441 | 1151 | 776 | 360 | 63 | 43 | 5235 |
| CA | 98 | 0 | 869 | 572 | 189 | 229 | 16 | 48 | 2021 |
| ESD | 749 | 2 | 3325 | 2018 | 3105 | 683 | 425 | 129 | 10436 |
| FMN | 899 | 8 | 4283 | 2242 | 3548 | 648 | 407 | 115 | 12150 |
| UCE | 500 | 1 | 2695 | 1626 | 1926 | 617 | 261 | 78 | 7704 |
| SalUn | 66 | 1 | 556 | 286 | 346 | 187 | 73 | 41 | 1556 |
| Ours | 43 ↓95.0% | 0 ↓100.0% | 486 ↓87.3% | 384 ↓81.1% | 132 ↓96.1% | 201 ↓70.5% | 7 ↓98.3% | 13 ↓89.4% | 1266 ↓88.8% |

