# OpenReview forum: "Dark Miner: Defend against unsafe generation for text-to-image diffusion models"
_ICLR.cc/2025/Conference — ICLR 2025 Conference Withdrawn Submission_

### Official Review · Reviewer_TYdk · 2024-11-02

**Soundness:** 2
**Presentation:** 3
**Contribution:** 2
**Rating:** 5
**Confidence:** 4

**Summary:**

This paper focuses on the model editing task of the text-to-image diffusion model. To address the challenges in erasing all unsafe prompts, this paper proposes a method called Dark MINER. This method consists of three steps: 1) mining the potential embeddings related to the unsafe images, 2) assessing whether the potential embeddings effectively induce the model to generate unsafe images, 3) if the mined embedding is effective, this paper conducts the erasing process. In step 3, to protect the generation of safe concepts, this paper also incorporates the regularization in three kinds of concepts: a predefined anchor concept, a null concept, and a concept '-c' that is defined as 'unsafe embedding * -1'.

**Strengths:**

- The paper presents a clear logical flow and is well-written.
- Experiments demonstrate the effectiveness of the proposed method.

**Weaknesses:**

1. Please clarify the difference from below existed studies [1,2].

2. In Table.1, the attack success rate of UnlearnDiff in the Violence concept is still 79%. Please provide an explanation for this. Similar phenomenon appear in P4D with the Violence concept (ASR is 46%) and the Church concept (ASR is 49).

3. In Eq.8, this paper proposes three regularization terms to protect the generation of safe concepts, but lacks of related ablation experiments to assess the effect of these three terms.

[1] RACE: Robust Adversarial Concept Erasure for Secure Text-to-Image Diffusion Model, ECCV 2024

[2] Reliable and Efficient Concept Erasure of Text-to-Image Diffusion Models, ECCV2024

**Questions:**

Please help to address weaknesses.

---

> ### Author Response · Authors · 2024-11-20
> **Official Response**
>
> We sincerely thank for your reviews. The followings are the responses to your concerns.
>
> > Weakness 1: Please clarify the difference from below existed studies [1,2].
> [1] RACE: Robust Adversarial Concept Erasure for Secure Text-to-Image Diffusion Model, ECCV 2024
> [2] Reliable and Efficient Concept Erasure of Text-to-Image Diffusion Models, ECCV2024
>
> Thanks for your suggestion. The method proposed in [1] is RACE and the method proposed in [2] is RECE.
>
> From the perspective of the motivations, RACE addresses the robustness of erasure, RECE addresses the incomplete erasure of a text, while our method addresses the incomplete erasure of a concept. The motivation of RACE stems from the fact that the erased content can be generated again by applying a small perturbation to an embedding of a text erased in the training. The motivation of RECE stems from the fact that texts that have been erased during the training phase can still generate relevant content. The focus of this paper is on the incomplete erasure of a concept, which refers to the fact that the limited texts in the training phase make it impossible to completely prevent the generation of diverse image contents related to concepts.
>
> From the perspective of the methods, RACE uses adversarial training, adding a small perturbation term to the embeddings during training. RECE derives the attention projection of text embeddings in the previous training epochs with the fine-tuned parameters and erases it in the current epoch. On the contrary, our method uses concept-related images to continuously mine the embeddings of concepts in the generative models, and then erases the mined embeddings.
>
> From the perspective of the experiments, our study conducts more detailed experiments. Specifically, in terms of baselines, our study compares six previous methods but RACE compares only one method and RECE compares five methods. In terms of attacks, our study involves four attacks while both RACE and RECE involve three attacks. Also, they evaluate the anti-attack performance on parts of concepts while we evaluate it on all the concepts we erase.
>
>
> > Weakness 2: In Table.1, the attack success rate of UnlearnDiff in the Violence concept is still 79%. Please provide an explanation for this. Similar phenomenon appear in P4D with the Violence concept (ASR is 46%) and the Church concept (ASR is 49).
>
> Thanks for your question. The violence is a concept which has diverse meanings. For example, a picture of a fight is violent, a picture of guns is violent, and a picture of blood is also violent. The diverse connotations of this concept make it difficult to completely erase it. This point also further illustrates that **simply collecting some texts is difficult to encompass the rich connotations of a concept**, which leads to the poor performance of previous methods. Our method aims to address this challenge by iteratively mining the representations of concepts contained in the model. From Table 1, we can see that the results of our method, especially the anti-attack performance, are significantly better than those of other methods. Similar situations exist for the church concept.
>
>
> > Weakness 3: In Eq.8, this paper proposes three regularization terms to protect the generation of safe concepts, but lacks of related ablation experiments to assess the effect of these three terms.
>
> Thanks for your suggestion very much. The ablation results are as follows. The experiment is conducted on the concept of nudity and the running epoch is set to 20.
>
> |$c_0$|$0c$|$-c$|Ratio↓|CLIP↑|
> |:---:|:---:|:---:|:---:|:---:|
> |×|√|√|17.64|30.26|
> |√|×|√|16.19|30.11|
> |√|√|×|15.97|30.09|
> |√|√|√|20.96|30.76|
>
>
> In the above table, $c_0$ denotes the anchor prompt *a photo*, $c$ denotes the mined embeddings, and $\gamma c$ represents the dot-product between a scalar $\gamma$ and $c$ ($\gamma =0$ or $\gamma=-1$). From the results, we can see that $-c$ is the most important preservation term. The reason is that $c$ is changing in the training process and $-c$ can help protect more irrelevant embeddings. It should be noted that the above experimental results are obtained at the 20th running epoch. As the training continues, the gap in the preservation performance will continue to increase. In addition, we also observe that removing some preservation terms leads to better erasure performance. This is because the erasure speed will be accelerated when the preservation is weakened. The erasure performance will be better when the model is evaluated at a certain number of epochs rather than at the end of training. We supplement these results in Section 4.3.

---

> ### Author Response · Authors · 2024-11-25
>
> Dear Reviewer,
>
> Thank you again for your valuable review! We have thoroughly addressed your concerns by elaborating on our work, conducting new experiments, and revising the paper. Could you please kindly review our responses and let us know if you have any further comments or suggestions?
>
> We would appreciate it if you reconsider your score in light of our new improvements made to the paper.
>
> Thank you for your time and consideration!

---

> ### Author Response · Authors · 2024-11-26
> **Request for discussion**
>
> Dear reviewer,
>
> With the extension of the discussion period, I am writing to kindly request your valuable feedback and reconsider your previous rating, if possible. Your insights would be greatly appreciated as they would help further refine our work.
>
> Thank you for your time and consideration.

---

> > ### Comment · Reviewer_TYdk · 2024-11-26
> >
> > I do not have furthermore question, but this ablation study seems some extra studies and efforts need to apply.  I would like to maintain my score.

---

> > > ### Author Response · Authors · 2024-11-26
> > >
> > > Thank you for your feedback. Could you please provide more specific details on what aspects require further investigation?

---

> > > ### Author Response · Authors · 2024-11-27
> > > **Request for discussion**
> > >
> > > Hi reviewer,
> > >
> > > As we are approaching the extended discussion phase, could you kindly engage in a discussion with us?

---

### Official Review · Reviewer_suXv · 2024-11-03

**Soundness:** 3
**Presentation:** 2
**Contribution:** 3
**Rating:** 5
**Confidence:** 4

**Summary:**

This work proposes, Dark Miner, an approach designed to address unsafe content generation in text-to-image diffusion models. Unlike existing methods that mainly adjust generation probabilities for known unsafe textual inputs, Dark Miner emphasizes minimizing unsafe generation probabilities for unseen or adversarial prompts. This is achieved through a recurring three-stage process: mining embeddings with high probabilities for unsafe content, verifying them, and circumventing unsafe generation pathways. Some experimental results demonstrate that Dark Miner outperforms six state-of-the-art methods in erasing and defending against unsafe content.

**Strengths:**

1. The research topic in this paper is relevant to the community.
2. Organization of the paper is relatively clear, even not perfect.
3. Experimental details are clearly stated.

**Weaknesses:**

1. The author claims that the following as one of the contributions:
“Based on the total probabilities, we analyze the reason why existing methods cannot completely erase concepts for text-to-image diffusion models and are vulnerable to attacks.”
I did not find a (sub)section for this part, although some discussion can be found in some paragraphs.
The reason cannot be found in conclusion section either.

2. Missing related work: The proposed Dark Miner aims to emphasize unsafe generation for unseen or adversarial prompts. One of the previous work [1] also handle unseen or adversarial harmful prompts via building a robust detection space, which is missing in the related work.

[1] Liu, Runtao, et al. "Latent guard: a safety framework for text-to-image generation." ECCV, 2024.

3. Experiments: Previous work show their effectiveness on many harmful concepts. This work only conducts experiments on two. The generalisation of the proposed approach remain to be further verified. The previous approach also shows very different formance on different harmful concepts.

4. The proposed approach cannot handle prompts to generate biased images, e.g. gender bias? Bias is also a harmful concept in responsible text-to-image generation [2,3].

[2] Li, Hang, et al. "Self-discovering interpretable diffusion latent directions for responsible text-to-image generation." Proceedings of the IEEE/CVF Conference on Computer Vision and Pattern Recognition. 2024.
[3] Friedrich, Felix, et al. "Fair diffusion: Instructing text-to-image generation models on fairness." arXiv preprint arXiv:2302.10893 (2023).

Minor issues:
- The citation format is incorrect across the paper.
- Confusing annotation, e.g. 0c in Equation 8

**Questions:**

1. The paper claims that existing methods fail to guarantee safe generation for unseen texts in the training phase. Does the proposed Dark Miner can provide such guarantee? If so please provide more details or discussion regarding this.

2. In the mining stage of DM2, can DM2 obtain the novel harmful prompts? E.g. this work shows that text-to-image models also suffer from Multimodal Pragmatic Jailbreak prompts [1]. Can the Jailbreak prompts also be learned by the proposed DM2? Some discussion about the limitation of the mining stage will be helpful to the readers.

[1] Liu, Tong, et al. "Multimodal Pragmatic Jailbreak on Text-to-image Models." arXiv preprint arXiv:2409.19149 (2024).

3. How the proposed approach performs if the system is black-box? Is it still feasible?

---

> ### Author Response · Authors · 2024-11-20
> **Official Response to Weaknesses**
>
> We sincerely appreciate your reviews. The followings are the responses to your concerns.
>
> > Weakness 1: The author claims that the following as one of the contributions: “Based on the total probabilities, we analyze the reason why existing methods cannot completely erase concepts for text-to-image diffusion models and are vulnerable to attacks.” I did not find a (sub)section for this part, although some discussion can be found in some paragraphs. The reason cannot be found in conclusion section either.
>
> We have presented it in Section 3.1.
>
> Specifically, we decompose the concept generation probability into the conditional probability of the texts, and then illustrate the shortcoming of existing methods that they cannot ensure the minimization of concept generation probability due to limited collected texts.
>
> > Weakness 2: Missing related work: The proposed Dark Miner aims to emphasize unsafe generation for unseen or adversarial prompts. One of the previous work [1] also handle unseen or adversarial harmful prompts via building a robust detection space, which is missing in the related work.
> [1] Liu, Runtao, et al. "Latent guard: a safety framework for text-to-image generation." ECCV, 2024.
>
> Thanks for your suggestion. Since Latent Guard uses a large amount of inappropriate texts to train the model, we use the official pre-trained model to conduct experiments on the erasure of nudity and violence. Other configurations follow the ones reported in the paper. The results are as follows. The results show that the erasure performance of our method, including the evaluation on general prompts (Ratio) and adversarial prompts (RAB), outperforms the one of Latent Guard significantly while our generative capability is also superior to the one of Latent Guard (CLIP and FID).
>
> |Method (Nudity)|Ratio↓|RAB↓|CLIP ↑|FID ↓|
> |:----------:|:------------:|:-------------:|:-------------:|:-------------:|
> |Latent Guard |36.3|36.1|29.0|24.9|
> |Ours|**12.1**|**26.2**|**30.0**|**21.7**|
>
>
> |Method (Violence)|Ratio↓|RAB↓|CLIP ↑|FID ↓|
> |:----------:|:------------:|:-------------:|:-------------:|:-------------:|
> |Latent Guard |32.0|45.2|29.0|24.9|
> |Ours|**19.1**|**35.1**|**29.5**|**22.4**|
>
> > Weakness 3: Experiments: Previous work show their effectiveness on many harmful concepts. This work only conducts experiments on two. The generalisation of the proposed approach remain to be further verified. The previous approach also shows very different formance on different harmful concepts.
>
> We have presented the performance of other five harmful concepts in Appendix A, as mentioned in Line 361 in Section 4.2.
>
> From the results, we can see that our method achieves the lowest harm ratio on all other concepts. Our method also has competitive effectiveness on erasing these harmful concepts.
>
> > Weakness 4: The proposed approach cannot handle prompts to generate biased images, e.g. gender bias? Bias is also a harmful concept in responsible text-to-image generation [2,3].
> [2] Li, Hang, et al. "Self-discovering interpretable diffusion latent directions for responsible text-to-image generation." Proceedings of the IEEE/CVF Conference on Computer Vision and Pattern Recognition. 2024.
> [3] Friedrich, Felix, et al. "Fair diffusion: Instructing text-to-image generation models on fairness." arXiv preprint arXiv:2302.10893 (2023).
>
> The focus of this paper is on the concept erasure task, with an emphasis on studying how to erase certain concepts in generative models so that they can no longer generate these concepts. Bias correction is out of the scope of this paper, and responsible generation is also a very broad research topic.
>
> The early and later related works, such as SLD, ESD, SalUn, and Latent Guard, all do not consider bias correction. Furthermore, the references [2,3] do not consider the objects and the styles, the two concepts which are always considered in the task of concept erasure on the contrary.
>
> > Weakness 5: Minor issues:The citation format is incorrect across the paper.Confusing annotation, e.g. 0c in Equation 8.
>
> Thanks for your careful review and we correct the citation in the revision.
>
> For Eq.8, as mentioned in Line 261 in Section 3.2.2, $\gamma c$ represents a dot-product between the scalar $\gamma$ and the vector $c$.  It is used to provide diverse embeddings for preserving in the training.

---

> ### Author Response · Authors · 2024-11-20
> **Official Response to Questions**
>
> > Question 1: The paper claims that existing methods fail to guarantee safe generation for unseen texts in the training phase. Does the proposed Dark Miner can provide such guarantee? If so please provide more details or discussion regarding this.
>
> Thanks for your question. The demonstration of this point is implicit in our experimental design. In the experiments, neither the user prompts used to verify the erasure performance nor the adversarial prompts used to verify the anti-attack performance are involved in the training of any methods. Therefore, the results can demonstrate the ability of these methods to guarantee safe generation for texts that are not seen during training. We supplement this discussion in Section 4.2.
>
> > Question 2: In the mining stage of DM2, can DM2 obtain the novel harmful prompts? E.g. this work shows that text-to-image models also suffer from Multimodal Pragmatic Jailbreak prompts [1]. Can the Jailbreak prompts also be learned by the proposed DM2? Some discussion about the limitation of the mining stage will be helpful to the readers.
> [1] Liu, Tong, et al. "Multimodal Pragmatic Jailbreak on Text-to-image Models." arXiv preprint arXiv:2409.19149 (2024).
>
> Thanks for your question. Multimodal Pragmatic Jailbreak is a novel work released in September recently. It discusses prompts that can generate visual texts, which refer to the texts in images. We try one example mentioned in this paper. Specifically, we sample some images using the prompt "An image, with a sign that says 'sea monkeys'.'' Using SD v2.1, we perform the mining stage. The images generated with the mined embeddings are shown in **Appendix I**. It should be noted that the generation of visual texts is always a challenging problem for text-to-image diffusion models, so the visual texts generated by SD v2.1 often contain errors. We can see that our method can mine the prompts corresponding to these visual texts.
>
> Actually, what concepts to be mined depends on how we configure the image pool. The images contained in the image pool embody the concept of what we want to erase. If we include the images containing visual texts in the image pool, our mining can learn the common representation of these images. This principle is similar to Textual Inversion [1]. If the image pool contains images that are irrelevant to the target concepts, the mining and erasure performance will be affected.
>
> [1] Rinon Gal, et al. An image is worth one word: Personalizing text-to-image generation using textual inversion. ICLR, 2022.
>
> > Question 3: How the proposed approach perform if the system is black-box? Is it still feasible?
>
> We believe that the erasing methods, including previous methods and our method, are not suitable for black box systems. These methods all prevent concept generation by changing model parameters or the generation process. In the setting of the black box, we cannot obtain the model parameters or structures, so there is no way to intervene in the model training, fine-tuning, or generation process.

---

> ### Author Response · Authors · 2024-11-25
>
> Dear Reviewer,
>
> Thank you again for your valuable review! We have thoroughly addressed your concerns by elaborating on our work, conducting new experiments, and revising the paper. Could you please kindly review our responses and let us know if you have any further comments or suggestions?
>
> We would appreciate it if you reconsider your score in light of our new improvements made to the paper.
>
> Thank you for your time and consideration!

---

> ### Author Response · Authors · 2024-11-26
> **Request for discussion**
>
> Dear reviewer,
>
> With the extension of the discussion period, I am writing to kindly request your valuable feedback and reconsider your previous rating, if possible. Your insights would be greatly appreciated as they would help further refine our work.
>
> Thank you for your time and consideration.

---

> > ### Comment · Reviewer_suXv · 2024-11-26
> >
> > Thank authors for their efforts during the rebuttal process. Many of my concerns have been well addressed.
> >
> > However, I still encourage the authors to discuss the limitations of their approach from the perspective of bias. While I understand that previous works primarily focused on handling harmful concepts, ethical concerns in practical image generation applications extend beyond that. Notably, a recent CVPR 2024 publication [1] demonstrates the ability to address both harmful concepts and biases simultaneously. Including a discussion on the limitations of this work in handling bias would provide valuable insight to the community and help guide follow-up research toward extending this work into a more general framework.
> >
> > I also checked Section 3.1 for the first contribution before writing my comments. It is just a simple reformulation. I still do not get what are the deep reason why existing methods cannot completely erase concepts for text-to-image diffusion models and are vulnerable to attacks. I think think more details would be helpful for more boarder audience.
> >
> > I also read other review comments. I am will raise my score if the concerns can be addressed.
> >
> > A reminder: you can update your paper if you like to include any discussion/exps/analysis in your submission, if I understand the system correctly.
> >
> > [1] Li, Hang, et al. "Self-discovering interpretable diffusion latent directions for responsible text-to-image generation."

---

> > > ### Author Response · Authors · 2024-11-26
> > >
> > > Thank you very much for your valuable feedback.
> > >
> > > For the task of debiasing, it is a potential limitation of our method at present. This paper mainly focuses on mining and erasing concepts, and cannot yet correct biased concepts. One possible approach is to learn the representations of professional concepts in the mining stage and then align the generation probability of these concepts in different social groups, but this needs to be confirmed by further experiments. We have supplemented this discussion in Appendix G.2.
> > >
> > > [Modification in Appendix G.2]
> > > > G.2 APPLICATION TO DEBIASING
> > > Given that recent works (Li et al., 2024; Friedrich et al., 2023) have delved into the bias issue in text-to-image generative models in addition to addressing concept erasure, the performance of our method on the debiasing task needs further verification. This constitutes a limitation of the present study, as our paper primarily focuses on mining and erasing concepts. A potential pathway to leveraging our method for debiasing is to learn the representations of professional concepts in the mining stage and then align the generation probability of these concepts across various social groups. We defer the conduct of related experiments to future work.
> > >
> > > For the first contribution, the reformulation is the foundation of our following analysis and our method. By this reformulation, we point out that the limitation of collected texts is the main shortcoming of previous methods. Specifically, these methods usually collect texts related to target concepts first and then modify their conditional generation probability. However, we cannot exhaust these relevant texts such as adversarial texts, leading to the generation of erased concepts conditioned on those texts which are not seen in the training. Not only previous methods, but also some recent methods such as Latent Guard [1] still show such shortcoming.
> > >
> > > [1] Liu, Runtao, et al. "Latent guard: a safety framework for text-to-image generation." ECCV, 2024.
> > >
> > > Further, in Section 3.2, we derive a tight upper bound of the generation probability of a target concept, and propose our method to minimize this upper bound so that the generation of the concept can be reduced. In fact, this is a process of finding the optimal embeddings of target concepts, instead of manually collecting the embeddings of the related texts.
> > >
> > > The latest revision has been uploaded, with the previous and new revisions highlighted. Limited by space, some discussions/experiments/analyses are placed in the Appendix. Thank you very much for your careful reminder and we are looking forward to your reply.

---

> > > ### Author Response · Authors · 2024-12-01
> > >
> > > Dear Reviewer,
> > >
> > > As we are approaching the deadline of the discussion, I kindly inquire whether the above response has addressed your concerns and whether you might reconsider your rating in light of this information.
> > >
> > > Best,
> > >
> > > Submission5485 Authors

---

### Official Review · Reviewer_oRfd · 2024-11-03

**Soundness:** 3
**Presentation:** 3
**Contribution:** 2
**Rating:** 5
**Confidence:** 4

**Summary:**

This paper proposed Dark Miner to eliminate unsafe content generation of T2I models. It searches and verifies the embeddings which contain the unsafe concept and reduces unsafe generation by adjusting LoRA adapter of T2I models.
The paper mainly made following contributions:
1.	Point out the previous methods fail to avoid unsafe generation on out-of-distribution prompts and easily being tricked by attacking methods.
2.	Propose Dark Miner which includes three stage to reduce the optimal embeddings related to unsafe concept and maintain generation ability on benign prompts.
3.	Evaluate efficient of proposed compared with 6 SOTA methods and conduct 4 SOTA attacks.

**Strengths:**

Authors point out the limitation of existing method and provide theoretical analysis.
Authors propose an iterative scheme to mine the text c that with maximum likelihood to generate unsafe content while previous method usually predefined such text c.
Authors propose a method to avoid overly concept removal by verifying the embedding before circumventing. Authors apply CLIP model to extract delta features from reference image and generated image, then a new metric based on cosine similarity of delta features.
Authors make effect to test proposed method against 4 different attacks.

**Weaknesses:**

In section 3.2.2, authors use prompt “a photo” to generate reference image and compare the distance with target image generated by mined embedding.  However, due to the randomness of reference image, the difference between reference image and target image is always high. In an extreme case, when there are benign images or non-explicit sexual related images in the image pool, the verifying and circumventing will be affected.

And the prompt of target image has no relation with prompt of reference image. Another potential drawback is that, all the concepts from “unsafe” prompt are shifted away to random direction without guidance. Therefore, the image generated by “unsafe” prompt will not maintain any semantic information from its prompt even there are safe content expressed by “unsafe” prompt. The generation will be random which might degrade the utility of model.

In evaluation metrics, author use the mean classification score to evaluate inappropriateness by NudeNet, however, author didn’t mention what classes and threshold used in their implementation. And it is better if authors could also show the number of images being classified as inappropriate image instead of classification score.

For the CLIP score, the proposed method has relative low performance compared with other SOTA methods.

**Questions:**

What classes are used in NudeNet classifier and threshold?
SLD and ESD original paper test their FID score on COCO-30k which is from COCO 2024 validation dataset, why authors didn’t use the same one?
According to another relevant paper in the task of nudity elimination as following: Li, Xinfeng, et al. "SafeGen: Mitigating Unsafe Content Generation in Text-to-Image Models." arXiv preprint arXiv:2404.06666 (2024). In the Table 3, they also give the FID score on COCO 2017 validation dataset, achieving 20.36 and 20.31 on ESD and their proposed method SafeGen, which is better than the author’s implementation. We suggest author should revisit their FID calculation on ESD on COCO 2017.

---

> ### Author Response · Authors · 2024-11-20
> **Official Response to Weakness 1 and 2**
>
> We sincerely appreciate your reviews. The followings are the responses to your concerns.
>
> > Weakness 1: In section 3.2.2, authors use prompt “a photo” to generate reference image and compare the distance with target image generated by mined embedding. However, due to the randomness of reference image, the difference between reference image and target image is always high. In an extreme case, when there are benign images or non-explicit sexual related images in the image pool, the verifying and circumventing will be affected.
>
> Before addressing your concern, I'd like to clarify one detail first. As mentioned in Section 3.2.2 and Appendix E, we do not directly compare the distance between reference images and target images. We calculate **the difference feature between a target image and a reference image**, and **the difference feature between an image in the pool and this reference image**. The cosine similarity between these two difference features measures the unsafe level of the mined embedding. This is to say, we use the direction of the difference feature vectors to represent the similarity of the image concepts.
>
> To evaluate the effectiveness of the verifying step, we conduct a classification experiment. Specifically, using SD v1.4, we sample 100 images using the prompt "a photo" as the reference images, and 100 images using the prompt "a photo of [CONCEPT]" as the images in the image pool. Then, for each concept, the images with the concept in the image pool and the reference images without the concept are regarded as the "positive" and "negative" classes respectively, and we calculate the proposed metrics for these images. In total, there are 2\*100\*100\*100=2,000,000 pairs of samples (100 reference images, 100 images in the pool, and 100 images with/without the concept). We use these sample pairs to calculate AUC scores. The results are as follows. The results demonstrate that our method can help identify images effectively. We supplement these results in Appendix E.2.
>
> |Concept|AUC|
> |---|---|
> |Nudity| 0.990 |
> |Violence | 0.975 |
> |Church | 0.989 |
> |French horn | 1.000 |
> |Van Gogh's painting style| 0.997 |
> |Crayon painting style | 0.960 |
>
>
> Last but not least, the image pool **should contain images with the concepts that we want to erase**. For example, if we want to erase nudity, we should ensure that the image pool includes related images, rather than benign or non-explicit images. This is analogous to the previous method, where we should gather text related to nudity, excluding irrelevant text.
>
>
> > Weakness 2: And the prompt of target image has no relation with prompt of reference image. Another potential drawback is that, all the concepts from “unsafe” prompt are shifted away to random direction without guidance. Therefore, the image generated by “unsafe” prompt will not maintain any semantic information from its prompt even there are safe content expressed by “unsafe” prompt. The generation will be random which might degrade the utility of model.
>
> Your concern comes from the possibility that our method degrades the ability to generate the safe part of an unsafe prompt. We reduce the negative impact in two ways. On the one hand, We improve **the purity of optimized embeddings**. In the mining step, multiple images are used to optimize an embedding. The target concept is the common content in these images. The embedding is optimized to learn the cross-image concept representations so that the optimized embeddings can exclude irrelevant content as much as possible. On the other hand, we improve **the diversity of the embeddings that are preserved**. In the circumventing step, $-c$ is preserved. Because the mined embedding $c$ changes in every loop and the circumventing step samples different embeddings for each iteration, the preserved embeddings are diverse.

---

> > ### Comment · Reviewer_oRfd · 2024-11-25
> >
> > How do you define “how much” differences you need between these two feature vectors to ensure the safe content generation?  In terms of nudity erasing, generated image can have high similarity with the reference image but actually without any nudity. How your method behaves in this situation?
> >
> > For the nudity concept, how do you acquire nudity images for image pool? What’s the impact when you have limited access to nudity images? Will model’s erasing performance rely on what you have in your image pool (e.g., only cartoon style or painting style nudity in image pool)?

---

> > > ### Author Response · Authors · 2024-11-26
> > >
> > > Thank you very much for your feedback.
> > >
> > > For the first question, this situation is theoretically unlikely and has not been empirically observed.
> > >
> > > The reference images you mention are generated using the prompt *a sexual photo*. These images themselves have a strong diversity. In the mining stage, we use multiple images to optimize embeddings, making embeddings contain the common elements between these images (i.e. nudity) but the similarity with each of them is low. Although CLIP does not have the ability to identify fine-grained features between similar images, it can effectively help determine the erasure process under our experimental implementations, as demonstrated in the response to your weakness 1.
> > >
> > > For the second question, we form the pool by generating images using the prompt *a sexual photo*. It can lead to various nudity images in SD v1.4. If there are too few images, the performance will be affected. However, this does not mean that increasing the number of images will continue to improve the performance, because some images have not been sampled and the training stops. In Tab.3, we conduct an experiment on nudity with different sizes of the pool. The erasure performance is 21.5\% when the number of images is 20, 12.1\% when the number of images is 200, and 11.8\% when the number of images is 2000.
> > >
> > > For your last question, we try to generate some nudity images in the cartoon style but it fails. The generated images are in the cartoon style but have a low level of nudity. This prevents us from verifying the erasure performance under cartoon pictures yet. But we believe that the composition of the pool should reflect the erasure demands and include content related to what we want to erase. If only cartoon nudity is included, we indicate that we want to erase cartoon nudity. This may be an advantage of our method, as it may be possible to erase cartoon nudity while retaining real nudity, but further comparison and verification are needed.
> > >
> > > We are looking forward to your reply.

---

> > > ### Author Response · Authors · 2024-11-28
> > >
> > > Dear Reviewer,
> > >
> > > I hope this message finds you well.
> > >
> > > I appreciate the additional questions and insights you provided in response to my earlier answers. In turn, I have submitted my further responses addressing those concerns. However, I have not received any further feedback from you since then. Could you kindly review my latest responses and let me know if there are any remaining issues or if further clarification is needed?
> > >
> > > Thank you for your time and consideration.
> > >
> > > Best

---

> ### Author Response · Authors · 2024-11-20
> **Official Response to Weakness 3 and 4**
>
> > Weakness 3: In evaluation metrics, author use the mean classification score to evaluate inappropriateness by NudeNet, however, author didn’t mention what classes and threshold used in their implementation. And it is better if authors could also show the number of images being classified as inappropriate image instead of classification score.
>
> Thanks for your suggestions! The classes in the evaluation include all the exposed classes except exposed feet. The threshold is 0.5. The corresponding numbers of images in these classes are as follows. These details and the results are supplemented in Section 4.1 and Appendix J.
>
> |Method|Buttock|Anus|Armpits|Belly|Female Breast|Male Breast|Female Genitalia |Male Genitalia|Total|
> |------|:------:|:------:|:------:|:------:|:------:|:------:|:------:|:------:|:------:|
> |SD|856 | 4 | 3838 | 2035 | 3340 | 681 | 410 | 123| 11287 |
> |SLD|401|0|2441|1151|776|360|63|43|5235|
> |CA|98|0|869|572|189|229|16|48|2021|
> |ESD|749|2|3325|2018|3105|683|425|129|10436|
> |FMN|899|8|4283|2242|3548|648|407|115|12150|
> |UCE|500|1|2695|1626|1926|617|261|78|7704|
> |SalUn|66|1|556|286|346|187|73|41|1556|
> |Ours|43 (↓94.98%)|0 (↓100.0%)|486 (↓87.34%)|384 (↓81.13%)|132 (↓96.05%)|201 (↓70.48%)|7 (↓98.29%)|13 (↓89.43%)|1266 (↓88.78%)|
>
> > Weakness 4: For the CLIP score, the proposed method has relative low performance compared with other SOTA methods.
>
> Although our method is slightly lower than some methods in CLIP score, the erasure performance is significantly better than these methods. In terms of the erasure performance, the method closest to ours is SalUn. However, both its CLIP Score and its FID Score are degraded significantly. In Section 4.2, we discuss this point. We also give some examples using COCO prompts in Fig.3. It can be seen that the images generated by SalUn miss many key elements in the prompts but our method preserves them.

---

> ### Author Response · Authors · 2024-11-20
> **Official Response to Questions**
>
> > Question 1: What classes are used in NudeNet classifier and threshold? SLD and ESD original paper test their FID score on COCO-30k which is from COCO 2024 validation dataset, why authors didn’t use the same one? According to another relevant paper in the task of nudity elimination as following: Li, Xinfeng, et al. "SafeGen: Mitigating Unsafe Content Generation in Text-to-Image Models." arXiv preprint arXiv:2404.06666 (2024). In the Table 3, they also give the FID score on COCO 2017 validation dataset, achieving 20.36 and 20.31 on ESD and their proposed method SafeGen, which is better than the author’s implementation. We suggest author should revisit their FID calculation on ESD on COCO 2017.
>
> For the classes in NudeNet and the threshold, please refer to the response to Weakness 3.
>
> For the versions of COCO, we carefully read the main papers and the supplementary materials of SLD and ESD, but we do not find any mention of using COCO 2024. We also cannot retrieve the COCO 2024 dataset. Could you provide its reference? In addition, SLD is released at CVPR 2023 and ESD is released at ICCV 2023, so how do they use a dataset released in the future?
>
> For FID, although SafeGen uses the same dataset as ours, the number of test images is different. FID requires a comparison under exactly the same implementation. This is why we give the FID of the original SD model. We need to evaluate FID in the same table.

---

> > ### Comment · Reviewer_oRfd · 2024-11-25
> >
> > COCO dataset used in SLD and ESD is COCO 2014 validation subset (sorry for the typo in previous comment).

---

> ### Author Response · Authors · 2024-11-25
>
> Dear Reviewer,
>
> Thank you again for your valuable review! We have thoroughly addressed your concerns by elaborating on our work, conducting new experiments, and revising the paper. Could you please kindly review our responses and let us know if you have any further comments or suggestions?
>
> We would appreciate it if you reconsider your score in light of our new improvements made to the paper.
>
> Thank you for your time and consideration!

---

> ### Author Response · Authors · 2024-12-01
>
> Dear Reviewer,
>
> As we are approaching the deadline of the discussion, I kindly inquire whether the above response has addressed your concerns and whether you might reconsider your rating in light of this information.
>
> Best,
>
> Submission5485 Authors

---

### Official Review · Reviewer_N56b · 2024-11-04

**Soundness:** 1
**Presentation:** 2
**Contribution:** 2
**Rating:** 3
**Confidence:** 4

**Summary:**

This paper introduces Dark Miner for mitigating unsafe content generation in text-to-image diffusion models.  Dark Miner involves a recurring three-stage process of mining, verifying, and circumventing. It's designed to iteratively identify and suppress unsafe content generation. Comprehensive experiments demonstrate the superior performance of Dark Miner.

**Strengths:**

1. This paper studies a critical area. The capability of these text2image models enables unauthorized usage of these models to generate harmful or disturbing content.
2. Experiments exhibit good overall performance.

**Weaknesses:**

1. The performance of Dark Miner is largely limited by the image pool related to c. That is to say, a lot of text that leads to concept c, which is not linked to the images in the pool, will not identified by the mining step.
2. Questionable performance by CLIP in Section 3.2.2. Can you discuss how well CLIP performs this task, as this is critical to your methods, I believe this is an important part of the ablation study.
3. Also, CLIP is used in both the model to be erased (i.e. SD) and Section 3.2.2 to identify the concept. However, texts that circumvent the defense will lead to images that cannot be identified by the CLIP.

4. Methods are only evaluated on two SD models. Will this method generalize well beyond the SD families?
5. Lack of ablation on the three steps of Dark Miner. For instance,
    * how well does the method perform with and without the verifying step?
    * Ablation over the size of the image pool
    * Ablation over different parameters and configurations of optimizing for the embeddings.

**Questions:**

Please see the weakness part.

**Details Of Ethics Concerns:**

the first two step, i.e. the mining and verifying, can be directly applied to mine and search for text embedding that generates malicious contents.

---

> ### Author Response · Authors · 2024-11-20
> **Official Response to Weakness 1, 2, and 3**
>
> We sincerely appreciate your reviews. The following are the responses to your concerns.
>
> > Weakness 1: The performance of Dark Miner is largely limited by the image pool related to c. That is to say, a lot of text that leads to concept c, which is not linked to the images in the pool, will not identified by the mining step.
>
> First, our method does not rely on texts, therefore addressing the shortcomings of previous methods due to the use of texts. You mention that a lot of texts should be identified in the erasure. The previous methods collect related texts to fine-tune the model. But **how many texts should we collect?** And **how do we collect texts to cover these "a lot of texts"?** Texts are diverse and it is difficult to ensure that we collect enough texts. Many adversarial studies have demonstrated this point, such as Ring-A-Bell, CCE, Prompt4Debugging, and UnlearnDiff Attack in our reference list.
>
> In this paper, we aim to address this issue. Our method does not rely on texts. It learns the common representation of a concept in some images. In this way, we do not have to consider how to describe the concepts in an image.  We can erase concepts using the corresponding image content and do not need to worry about how many text descriptions there are for this image content. You mention that a lot of texts lead to concepts but are not linked to the images in the pool. Our method does not target specific texts for erasure, but rather mines diverse conceptual representations through diverse image combinations to achieve a more thorough erasure effect. **Our method does not rely on specific texts and avoids the difficulty of collecting sufficient texts**, which is the advantage of our method. We have analyzed this point in Section 3.1.
>
> Second, the results reported in the paper demonstrate the effectiveness of our method compared with previous methods, especially the results under the adversarial attacks. Therefore, we believe that the statement "the performance of Dark Miner is largely limited" lacks evidence.
>
>
> > Weakness 2: Questionable performance by CLIP in Section 3.2.2. Can you discuss how well CLIP performs this task, as this is critical to your methods, I believe this is an important part of the ablation study.
>
> Thanks for your suggestion. Using SD v1.4, we sample 100 images using the prompt "a photo", and 100 images using the prompt "a photo of [CONCEPT]". For each concept, we use each one of the former images as the reference image, and each one of the latter as the target image. Then these images with/without the concept are regarded as the "positive" and "negative" classes respectively, and we calculate the proposed metrics for these images. In total, there are 2\*100\*100\*100=2,000,000 pairs of samples. We use these sample pairs to calculate AUC scores. The results are as follows.
>
> |Concept|AUC|
> |---|---|
> |Nudity| 0.990 |
> |Violence | 0.975 |
> |Church | 0.989 |
> |French horn | 1.000 |
> |Van Gogh's painting style| 0.997 |
> |Crayon painting style | 0.960 |
>
> The results demonstrate that our method can help identify images effectively. We supplement these results in Appendix E.2.
>
> > Weakness 3: Also, CLIP is used in both the model to be erased (i.e. SD) and Section 3.2.2 to identify the concept. However, texts that circumvent the defense will lead to images that cannot be identified by the CLIP.
>
> We suspect you have misunderstood our method.
>
> Our method is only used to fine-tune the model but has no change to the inference stage. In our method, the CLIP image encoder is used in the **training** stage for determining whether to fine-tune. In the generative models, the CLIP text encoder is used in the **inference** stage for embedding prompts. The CLIP text encoder and image encoder work separately in different stages. We do not believe that "texts that circumvent the defense" in the reference stage lead to "images that cannot be identified" in the training stage.
>
> Further, we do not apply any defense on the text encoder for texts. Our paper aims to prevent models from generating undesired content rather than editing or filtering texts including their embeddings.
>
> If the above response does not address your concern, could you clarify your concern further? Or could you provide some references or evidence?

---

> ### Author Response · Authors · 2024-11-20
> **Official Response to Weakness 4 and 5**
>
> > Weakness 4: Methods are only evaluated on two SD models. Will this method generalize well beyond the SD families?
>
> We report the results on **three** SD models, i.e. SD v1.4 (the main experiments), SD v1.5, and SD v2.0 (Tab.2). We also try to implement our method on PixArt-$\alpha$-512 and find the mining process can end soon, and the attack CCE is used to verify this result (Appendix G).  The SD family is a widely adopted generative model in the area of concept erasure in text-to-image diffusion models, and we align with this established convention. In the future, we will explore more models with unsafe generation issues to verify our method.
>
> > Weakness 5: Lack of ablation on the three steps of Dark Miner. For instance, how well does the method perform with and without the verifying step? Ablation over the size of the image pool. Ablation over different parameters and configurations of optimizing for the embeddings.
>
> (1) The verifying step acts as a decision maker to determine whether to continue the fine-tuning. If we remove this step, we can also set a maximum number of epochs to stop it. Therefore, the ablation results of this step have been shown in Tab.S2 in Appendix D. In Tab.S2, we present the performance at different epochs.
>
> As for the mining step and the circumventing step, the mining step provides embeddings for fine-tuning and the circumventing step fine-tunes the model to erase concepts. Therefore, if we ablate these two steps, the method cannot work and we cannot get results.
>
> (2) Thanks for your suggestion. We conduct this ablation experiment on the concept of nudity and the results are as follows.
>
> |Size of Image Pool|Ratio $\downarrow$|
> |---|---|
> |20| 21.5 |
> |200 | 12.1 |
> |2000 | 11.8 |
>
> In the paper, the method runs 48 epochs when we erase nudity (Tab.S2) and the number of the totally sampled images is 48\*3=144. Therefore when we enlarge the size of the image pool, the improvement of the performance is limited. We supplement these results in Section 4.3 and Table 3.
>
> (3) In Section 4.3, we report the results using different image pools for optimizing embeddings. In Appendix D, we further show the ablation results using different embedding lengths. Could you further claim what ablation experiments on optimizing embeddings should we conduct and what your concerns could be addressed by them?

---

> ### Author Response · Authors · 2024-11-25
>
> Dear Reviewer,
>
> Thank you again for your valuable review! We have thoroughly addressed your concerns by elaborating on our work, conducting new experiments, and revising the paper. Could you please kindly review our responses and let us know if you have any further comments or suggestions?
>
> We would appreciate it if you reconsider your score in light of our new improvements made to the paper.
>
> Thank you for your time and consideration!

---

> ### Author Response · Authors · 2024-11-26
> **Request for discussion**
>
> Dear reviewer,
>
> With the extension of the discussion period, I am writing to kindly request your valuable feedback and reconsider your previous rating, if possible. Your insights would be greatly appreciated as they would help further refine our work.
>
> Thank you for your time and consideration.

---

> ### Author Response · Authors · 2024-11-27
> **Request for discussion**
>
> Hi reviewer,
>
> As we are approaching the extended discussion phase, could you kindly provide us with some feedback or engage in a discussion with us?

---

> ### Comment · Reviewer_N56b · 2024-11-28
> **Response to the rebuttal by the authors**
>
> > Limitation on the image pools.
>
> I acknowledge that Dark Miner alleviates the need for a group of texts associated with the concept c. Yet, a pool of images associated with concept c is still required. My question is that such a pool of images representing the concept c is not sufficient for the method to mine the concept c. I believe the same set of questions still remains with the image-based concept mining presented in the work. For instance, how many images should we collect? And how do we collect images to cover these "a lot of images"? Images are more diverse and even high dimensional, which makes it more difficult to ensure that we collect enough images.
>
> Also, due to the high-dimension nature of the conditional tensor, the tensor embedding derived from the mining step might not necessarily be associated with or generate the images containing concept c.
>
> Furthermore, the method is highly sensitive to images in the pool used to mine the concept embedding and any confounding may influence the mined embedding. This is due to the high-dimension nature and diverse information contained in the images. For instance, you want to remove the concept of sheep, yet every image in the pool has a sheep on the grass. Then how would the method know whether you want to mine the sheep concept or the grass concept? This problem makes collecting the right image pool even harder.
>
> ---
>
> > Suggestion on improving Section 3.2.2.
>
> I do misunderstand Section 3.2.2 due to the structure of the paper. I initially thought it was a verification of the faithfulness of the concept embedding with regard to the concept. I would suggest the authors merge or move Section 3.2.2 after Section 3.2.3 CIRCUMVENTING EMBEDDINGS as it's essentially the stopping criteria for removing the process.
>
> ---
>
> > Concern with larger pool size
>
> If the performance saturation is due to the early stopping. Will longer training time or lower threshold $\tau$ help? Since we still see an increase in performance with 2000 images in the pool compared to 200, one would suspect if you keep training on a larger pool size, it would be better.
> Also, the larger pool size leads to better performance echos my first question, i.e. the limited number of images in the pool still remains a problem to efficiently mine an accurate concept representation
>
> ---
>
> > Ablation study on the verifying step
>
> I would suggest you perform a search on the pre-defined number of iterations to erase and report the best number to compare with Dark Miner w/ the verifying step. The intuition is that you replace something trivial as the baseline to ablate the verifying step. However, as it's coming towards the end of the rebuttal, this experiment will not effect my score. However, I suggest the author add it for a comprehensive analysis on the method.

---

> > ### Author Response · Authors · 2024-11-28
> >
> > Thank you for your feedback.
> >
> > > Limitation on the image pools.
> >
> > First, there is a fundamental difference between texts and images. Images can directly present the appearance of a concept. For example, for a naked person, we can have various texts to describe it, but their visual appearance is consistent. This point allows images to express concepts more directly than texts.
> >
> > Second, there is a fundamental difference between using texts in previous methods and using images in our method. Previous methods only modify the generative distribution of collected texts, but we do not simply replace the generative distribution of texts with the generative distribution of images or the embeddings of images. We emphasize using these images to mine the model's intrinsic and concept-related knowledge. When the model has performed some erasures but still remembers concept-related knowledge, how should this knowledge be represented? Since images directly describe the visual appearance of concepts, these images help learn the expression of concepts by knowledge that has not been erased, thereby helping us continue to erase the model.
> >
> > Third, the aim of using multiple images in one mining process is to learn the common element in these images, and this common element just points to the target concepts. If we look at each image in isolation, they are of course different. But there are common features between these images, and these common features can be learned in optimization. For the example you gave, that is, sheep on the grass. Firstly, when we build the image pool, we will not use grass in the prompts, but only rely on *a photo of sheep* or simply *sheep* to generate images. Secondly, the generated images may contain grass, but its appearance is random since we do not specify it in the prompts, so that it is difficult to be a common element in multiple images at the same time. In fact, this is verified in the erasure of church, because the church is often associated with the lawn in the courtyard, but the images generated by the mined embeddings still take the church as the main object.
> >
> > Last but not least, in Tab.2, we erase nudity using the image pools which are generated by the different random seeds. The averages, standard deviations, and 95\% confidence intervals indicate the small sensitivity to the pools.
> >
> > In addition, the mined embeddings are erased after being verified, and the effectiveness of this verifying method is also verified in the supplemental experiment (Response to Weakness 2 and Tab.S7). Therefore, it can be guaranteed that the mined embeddings are associated with the concept.
> >
> >
> > > Concern with larger pool size
> >
> > We do not believe that increasing the size will continually improve the performance.
> >
> > From the results, the size of the pool has increased from 200 to 2000 (an increase by a factor of 9), but the erasure performance has only increased by 0.3 (about 0.025 of the original value). This small change is mainly due to the experimental deviation caused by the different images used for mining. The results cannot be exactly the same. The number of training iterations in these two settings is basically the same. This shows that continuously growing the size is not beneficial for significantly improving performance. When the image pool is very small, the performance is limited mainly due to the bias by the sampled images. The erased concept is nudity, which contains many body parts in addition to gender. A very small number of images will result in some body parts missing, resulting in incomplete erasure. Nevertheless, the erasure performance with only 20 images is still better than most methods from Tab.1.
> >
> > > Ablation study on the verifying step
> >
> > Thanks for your suggestion. In Tab.S2, we have reported the performance using different thresholds as well as their corresponding training iterations and we re-list it below. We think it may answer your concern to some extent. Increasing the number of training iterations or lowering the threshold can improve the erasure performance at present, and we will further look for the turning point of erasure performance in the future.
> >
> > | Thr | \# Iter | Ratio $\downarrow$|
> > |---|---|---|
> > |0.4|15|23.5|
> > |0.3|20|21.0|
> > |0.2|48|12.1|
> >
> > > Suggestion on improving Section 3.2.2.
> >
> > Thanks for your suggestion. As the PDF revision deadline has passed, we will take this suggestion into consideration in subsequent revisions.
> >
> > Thank you again and we are looking forward to your reply.

---

> > > ### Comment · Reviewer_N56b · 2024-12-02
> > > **Response to authors**
> > >
> > > > Limitation on the image pools.
> > >
> > > How can you be so sure the common part of the images in the pool points directly to the concept of interest? As I said, there will be confounding in the images, no matter how you collect the images, either by synthesis or by human efforts. Even with diffusion, images generated by the prompt "sheep" will likely to have grass in it (this is just an example, no need to argue with me on the "sheep" example.).
> > >
> > > > Larger pool size
> > >
> > > If a lower threshold leads to better performance, then adding more images in the pool do give better performance, right?
> > >
> > > I am confused about the reason why larger pool size saturates. Is it due to the small training iterations and high threshold or it's just saturates? You mentioned this in the first response but stated that the increase is caused by experimental deviation in the latest response.
> > >
> > > Overall, I don't see using images has a clear advantage over texts. On the contrary, texts are highly abstracted and may avoid many of the confounding issues (but still have some bias introduced by the occurrence of texts phrases).

---

> > > > ### Author Response · Authors · 2024-12-02
> > > >
> > > > Thank you very much for your response.
> > > >
> > > > The threshold is not a factor in the performance with different image pool sizes because we fixed it in all the experiments. Under this condition, the training stops after a similar number of epochs, resulting in many images in the pool not being sampled. As a result, a larger pool does not bring a significantly better performance. A lower threshold can improve the erasure performance but it also influences the generation performance. It depends on how we balance them.
> > > >
> > > > Again, thank you for your time and effort in the review and discussion with us.

---

> > > > > ### Comment · Reviewer_N56b · 2024-12-02
> > > > > **Response to the comments**
> > > > >
> > > > > If you want to perform an ablation study on the pool size, you need to find a way to control other conditions. In your case, you should not let the threshold and epoch influence your ablation study on pool size. For example, you can decrease all the thresholds and increase training epochs in all experiments of different pool sizes.
> > > > >
> > > > > Best

---

> > > > > > ### Author Response · Authors · 2024-12-03
> > > > > >
> > > > > > Thank you very much for your suggestion!
> > > > > >
> > > > > > Best

---

> ### Author Response · Authors · 2024-12-01
>
> Dear Reviewer,
>
> As we are approaching the deadline of the discussion, I kindly inquire whether the above response has addressed your concerns and whether you might reconsider your rating in light of this information.
>
> Best,
>
> Submission5485 Authors

---

### Author Response · Authors · 2024-11-20
**Global Response**

We sincerely appreciate the time and the reviews of our manuscript by all the reviewers.

In pursuit of safer T2I generation, we analyze the limitations and underlying issues of existing erasure methods in defending against attacks and propose a new framework to address these challenges. We thank the reviewers for their acknowledgment of our methods (suXv), experiments (oRfd, suXv), and writing (oRfd, TYdk).

We have thoroughly addressed all raised concerns in our rebuttal, and encourage you to review the individual responses for further details. In addition, the manuscript has been revised according to the suggestions of the reviewers, with clear markings indicating the revisions.

---

### Note · Authors · 2024-12-03

I have read and agree with the venue's withdrawal policy on behalf of myself and my co-authors.